# Atomic-level Ru-Ir mixing in rutile-type (RuIr)O$_2$ for efficient and durable oxygen evolution catalysis

Yeji Park ®[1,2,9], Ho Yeon Jang[3,9], Tae Kyung Lee[2,4,9], Taekyung Kim[5,9], Doyeop Kim ®[1], Dongjin Kim ®[1], Hionsuck Baik[5], Jinwon Choi[6,7], Taehyun Kwon ®[6,7] ✉, Sung Jong Yoo ®[2,8], Seoin Back ®[3] ✉ & Kwangyeol Lee ®[1] ✉

The success of proton exchange membrane water electrolysis (PEMWE) depends on active and robust electrocatalysts to facilitate oxygen evolution reaction (OER). Heteroatom-doped-RuO$_x$ has emerged as a promising electrocatalysts because heteroatoms suppress lattice oxygen participation in the OER, thereby preventing the destabilization of surface Ru and catalyst degradation. However, identifying suitable heteroatoms and achieving their atomic-scale coupling with Ru atoms are nontrivial tasks. Herein, to steer the reaction pathway away from the involvement of lattice oxygen, we integrate OER-active Ir atoms into the RuO$_2$ matrix, which maximizes the synergy between stable Ru and active Ir centers, by leveraging the changeable growth behavior of Ru/Ir atoms on lattice parameter-modulated templates. In PEMWE, the resulting (RuIr)O$_2$/C electrocatalysts demonstrate notable current density of 4.96 A cm$^{-2}$ and mass activity of 19.84 A mg$_{Ru+Ir}^{-1}$ at 2.0 V. In situ spectroscopic analysis and computational calculations highlight the importance of the synergistic coexistence of Ru/Ir-dual-OER-active sites for mitigating Ru dissolution via the optimization of the binding energy with oxygen intermediates and stabilization of Ru sites.

Renewable energy-powered proton exchange membrane water electrolysis (PEMWE) enables the cost-effective production of green hydrogen and thereby the establishment of a sustainable energy supply[1,2]. However, the large-scale deployment of PEMWE is hindered by the absence of efficient and durable electrocatalysts for the oxygen evolution reaction (OER)[3]. The commonly used IrO$_2$ does not satisfactorily accelerate this reaction and is unstable in acidic environments[4–6], whereas rutile-type RuO$_2$ exhibits optimal affinity for OER intermediates (O*, OH*, and OOH*) and a high initial catalytic activity for OER but is more susceptible to metal-ion leaching than IrO$_2$[7–9].

Depending on the crystallinity of RuO$_2$ and accessibility of its subsurface active sites, the RuO$_2$-catalyzed OER can proceed through the lattice oxygen oxidation mechanism (LOM) or adsorbate evolution

[1]Department of Chemistry and Research Institute for Natural Sciences, Korea University, Seoul, Republic of Korea. [2]Hydrogen Fuel Cell Research Center, Korea Institute of Science and Technology, Seoul, Republic of Korea. [3]Department of Chemical and Biomolecular Engineering, Institute of Emergent Materials, Sogang University, Seoul, Republic of Korea. [4]Department of Chemistry and Biological Engineering, Korea University, Seoul, Republic of Korea. [5]Korea Basic Science Institute (KBSI), Seoul, Republic of Korea. [6]Department of Chemistry, Incheon National University, Incheon, Republic of Korea. [7]Research Institute of Basic Sciences, Core Research Institute, Incheon National University, Incheon, Republic of Korea. [8]Division of Energy & Environment Technology, KIST school, University of Science and Technology (UST), Daejeon, Republic of Korea. [9]These authors contributed equally: Yeji Park, Ho Yeon Jang, Tae Kyung Lee, Taekyung Kim. ✉e-mail: thyunkwon@inu.ac.kr; ysj@kist.re.kr; sback@sogang.ac.kr; kylee1@korea.ac.kr

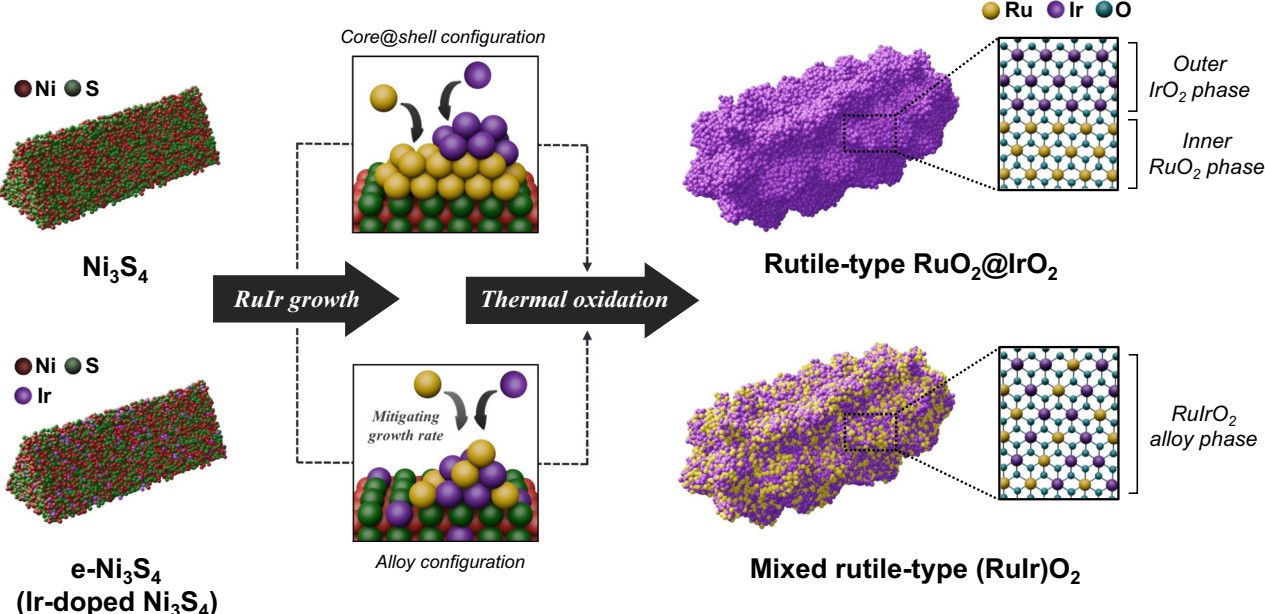

**Fig. 1 | Synthesis process of Ru/Ir oxide-based nanostructures.** Schematic illustration of Ru/Ir oxide-based nanostructures with different Ru/Ir atomic configurations. Red, green, yellow, purple, and cyan balls represent Ni, S, Ru, Ir, and O atoms, respectively.

mechanism (AEM)[10,11]. The kinetically favorable LOM is intrinsically detrimental to catalyst stability, as the facile overoxidation of exposed surface Ru species affords soluble $RuO_4^{2-}$ ions, thus accelerating catalyst degradation and reducing OER performance[12,13]. Consequently, the stability of $RuO_2$-based OER catalysts can be improved by suppressing the LOM and promoting the AEM.

The abovementioned mechanism steering can be achieved by doping $RuO_2$ with foreign elements (e.g., Pt), which share their electrons with the neighboring active Ru sites[14–16]. However, a major drawback of this approach is the inactivity of Pt atoms as OER catalytic sites. To solve this problem, we herein strategically placed electron-donating Ir atoms near active Ru sites to improve the stability of $RuO_2$-based OER catalysts and maximize the synergy between the two distinct OER active sites, namely Ru and Ir.

The Ir doping strategy relied on the formation of a well-mixed RuIr alloy, which, in turn, required the synchronous decomposition of Ru and Ir precursors. In typical solution-phase colloidal syntheses, the Ru precursor is reduced faster than the Ir precursor, which prevents the uniform distribution of Ru and Ir in the resulting catalysts. Given that the lattice mismatch between the template and growing metallic phase during template-mediated synthesis can affect the metal deposition rate[17–20], we hypothesized that the use of large-lattice-mismatch templates may decelerate the initial deposition of Ru and thus favor the formation of mixed RuIr alloy phases. To prove this idea, we prepared templates with different surface lattice parameters, namely pristine and lattice-expanded $Ni_3S_4$ nanorods, demonstrating that the abated deposition and attachment of Ru on the latter template resulted in the formation of a well-mixed RuIr phase.

The thermal oxidation of the above RuIr phase on lattice-expanded $Ni_3S_4$ (e-$Ni_3S_4$) afforded a well-mixed rutile-type (RuIr)$O_2$/C electrocatalyst with high OER performance. This catalyst showed a low overpotential of 174 mV at 10 mA cm$^{-2}$ and maintained its initial activity over 360 h of operation at a high current density of 100 mA cm$^{-2}$. When used as the anode catalyst layer of a PEMWE, (RuIr)$O_2$/C achieved a high current density of >4.96 A cm$^{-2}$ and mass activity of 19.84 A mg$_{Ru+Ir}^{-1}$ at 2.0 V, with minimal degradation observed over 250 h operation at 1.0 A cm$^{-2}$. In situ X-ray absorption spectroscopy (XAS) and in situ differential electrochemical mass spectrometry (DEMS) analyses, combined with density functional

theory (DFT) calculations, revealed that the incorporation of Ir atoms not only stabilized the local coordination environment around Ru, fostering the AEM pathway, but also facilitated Ir-to-Ru electron transfer via bridging oxygens, thereby hindering Ru dissolution during the OER. Furthermore, the coexistence of Ru and Ir at the cation sites of the rutile-type oxide phase at the atomic scale led to optimal oxygen-adsorbate binding energies and outstanding OER activity. Overall, the synergistic effect resulting from the atomic-level mixing of Ru and Ir effectively enhanced the activity and stability of the Ru-based OER catalyst under acidic conditions, providing valuable insights for the rational design of practical electrocatalysts. This study pave the way for the industrial-scale production of green hydrogen and thus contribute to establishing a sustainable society.

## Results
### Preparation of mixed rutile-type (RuIr)$O_2$/C
On the premise that the lattice parameter of the template surface is an important factor in the control of the atomic mixing between Ru and Ir atoms of the RuIr alloy phase (Fig. 1), we prepared the two types of templates: pristine $Ni_3S_4$ (Supplementary Fig. 1) and lattice-expanded $Ni_3S_4$ (e-$Ni_3S_4$) nanorods (Supplementary Fig. 2). The latter lattice-expanded template was prepared by introducing larger-sized Ir atoms (with a radius of 112 pm for Ir, compared to 110 pm for Ni) into the $Ni_3S_4$ framework. Notably, the powder X-ray diffraction (PXRD) pattern of e-$Ni_3S_4$ (Fig. 2a and Supplementary Fig. 2e) demonstrated a slight shift to a lower angle compared to $Ni_3S_4$, indicating the lattice expansion due to the embedding of Ir single atoms in the $Ni_3S_4$ matrix. The d-spacing analysis of $Ni_3S_4$ facets, derived from high-angle annular dark-field scanning transmission electron microscopy (HAADF-STEM) images and their corresponding fast Fourier transform (FFT) patterns (Fig. 2b and Supplementary Fig. 2d, f) revealed approximately 6% lattice expansion in the $Ni_3S_4$ phase of e-$Ni_3S_4$, providing compelling evidence of the structural modification achieved through Ir doping. Moreover, the X-ray photoelectron spectroscopy (XPS) analysis for Ni 2p and S 2p showed that the Ir dopant does not cause any changes in the electronic structure of the template but only alters the lattice structure (Supplementary Fig. 3 and Supplementary Tables 1, 2).

As anticipated, the Ru/Ir growth behavior depended on the lattice parameter of the template (Fig. 2c–j, Supplementary Fig. 4 and

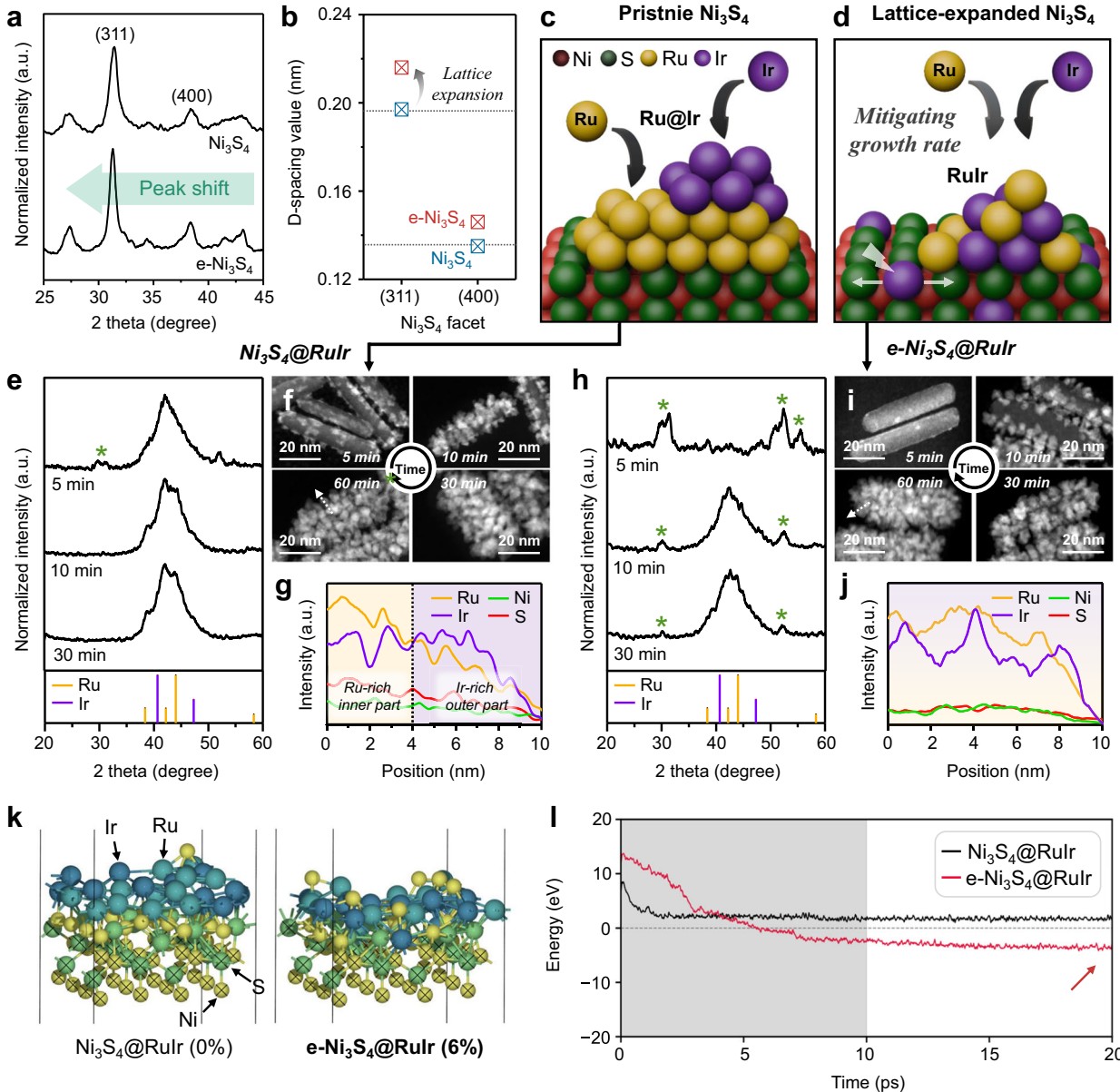

**Fig. 2 | Growth mechanism of Ru/Ir on two distinct templates. a** PXRD patterns of $Ni_3S_4$ and e- $Ni_3S_4$. **b** Comparison of the d-spacing values for the (311) and (400) facets of the $Ni_3S_4$ phase, observed in PXRD patterns in Fig. 2a. The two dashed lines represent the reference d-spacing values for the (311) and (400) facets. Schematic illustration of Ru and Ir growth on **c** $Ni_3S_4$ and **d** e-$Ni_3S_4$ templates, which afforded Ru@Ir and RuIr shell configurations, respectively. Green and red spheres denote Ni and S atoms. **e**, **h** PXRD patterns and **f**, **i** HAADF-STEM images (scale bar = 20 nm) of $Ni_3S_4$@RuIr and e- $Ni_3S_4$@RuIr obtained at reaction times of 5, 10, and 30 min.

Color bars and asterisks in PXRD patterns indicate the reference peaks of *hcp* Ru (yellow, #01-088-2333), *fcc* Ir (purple, #06-0598), and $Ni_3S_4$ (green, #01-076-1813). Line profile analysis of **g** $Ni_3S_4$@RuIr and **j** e-$Ni_3S_4$@RuIr determined along the lines marked by arrows in **f** and **i**, respectively. **k** Final images of the AIMD trajectories of $Ni_3S_4$@RuIr and e-$Ni_3S_4$@RuIr. Light green, yellow, turquoise, and navy spheres denote Ni, S, Ru, and Ir, respectively. **l** Energy profiles for the AIMD trajectories in **k**, with the gray region indicating the 10 ps equilibrium process.

Supplementary Note 1). For pristine $Ni_3S_4$, rapid Ru deposition followed by Ir deposition on the $Ni_3S_4$ was observed during the early stages of the reaction. This stepwise growth resulted in the formation of a Ru@Ir inner shell@outer shell configuration ($Ni_3S_4$@RuIr; Fig. 2g and Supplementary Fig. 5) However, when e-$Ni_3S_4$ was reacted simultaneously with $Ru^{3+}$ and $Ir^{3+}$, the initial rate of Ru deposition decreased owing to a significant lattice mismatch between the growing Ru phase and e-$Ni_3S_4$ surface. Consequently, the atomic-level mixing of Ru and Ir was promoted, leading to the formation of a RuIr alloy structure (e-$Ni_3S_4$@RuIr; Fig. 2j and Supplementary Fig. 6).

To understand the correlation between the lattice size of the templates and the growth behavior of Ru/Ir atoms on them, we

determined the average interatomic bond lengths ($d_{ave}$) in bulk and shell structures using DFT calculations (Supplementary Fig. 7, Supplementary Note 2, and Supplementary Data 1). When Ru atoms were grown on the templates, the difference in $d_{ave}$ between Ru bulk and e-$Ni_3S_4$@Ru (0.233 Å) was notably larger than that between bulk Ru and $Ni_3S_4$@Ru (0.098 Å), implying a lattice mismatch between the Ru atoms and e-$Ni_3S_4$ surface. In addition, the formation of Ru–Ru clusters upon the growth of Ru atoms on the e-$Ni_3S_4$ surface contributed to the decreased $d_{ave}$ of e-$Ni_3S_4$@Ru (Supplementary Fig. 8). Conversely, the difference in $d_{ave}$ between RuIr bulk and e-$Ni_3S_4$@Ru (0.121 Å) was similar to that between RuIr bulk and $Ni_3S_4$@Ru (0.144 Å), which suggested that the stability of the RuIr shell was maintained on the

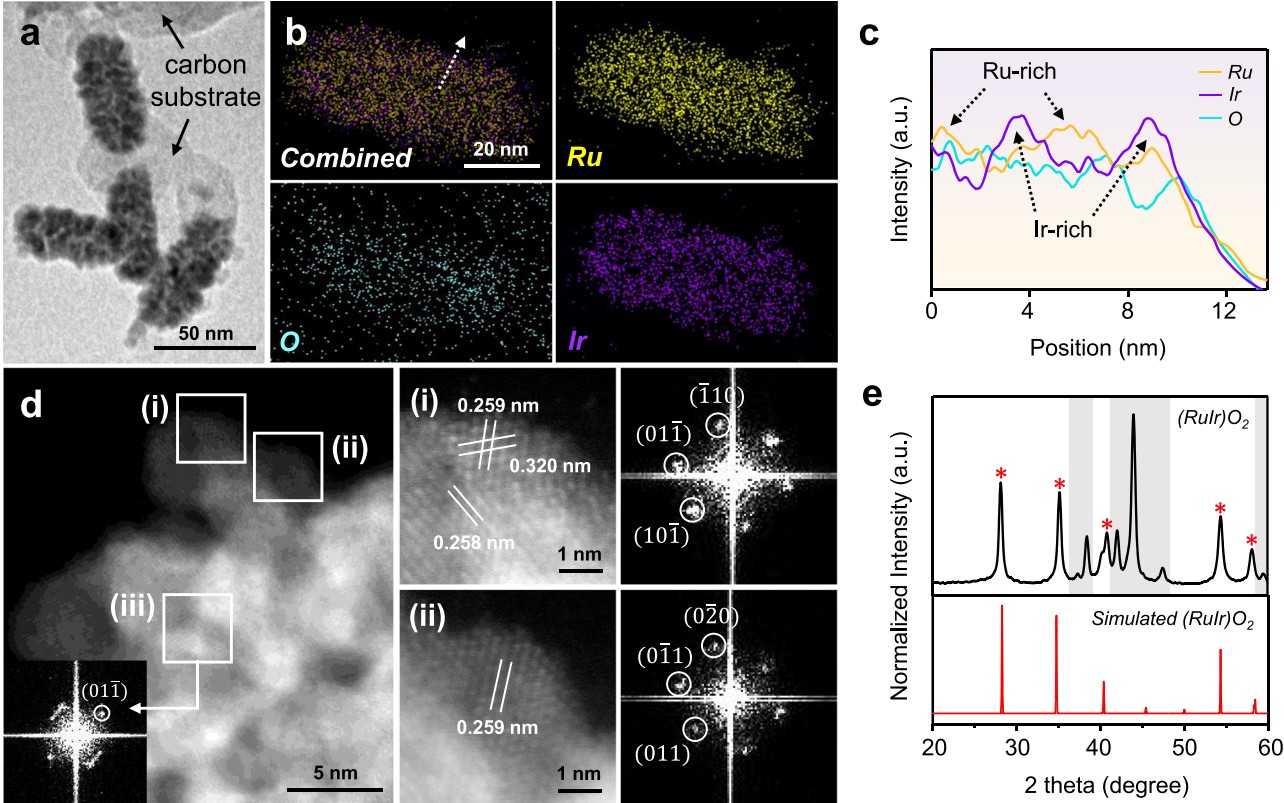

**Fig. 3 | Characterization of (RuIr)O₂/C electrocatalysts. a** TEM image of (RuIr)O₂/C prepared by the thermal oxidation of the RuIr phase on e-Ni₃S₄. **b** Combined and individual EDS elemental mapping images of Ru (yellow), Ir (purple), and O (cyan). The white arrow indicates the range of the line profile in **c**. **c** Line profile analysis of (RuIr)O₂/C corresponding to the marked area indicated by the white arrow in **b**. **d** Normal and enlarged HAADF-STEM images of (RuIr)O₂/C with corresponding FFT patterns. White boxes in **d** (i)–(ii) and (iii) indicate oxide and metallic species, respectively. **e** PXRD patterns of (RuIr)O₂/C. Gray boxes denote the remaining metallic species in (RuIr)O₂/C.

expanded core. Further Ab initio molecular dynamics (AIMD) simulations using a Ni₃S₄ core–metal shell interface model were conducted to rationalize the stabilization of the RuIr phase on the e-Ni₃S₄ surface (Fig. 2k, l). The RuIr shell structure was stabilized in the e-Ni₃S₄ model, which indicated that lattice expansion of the Ni₃S₄ phase promoted the simultaneous deposition of Ru and Ir atoms. Thus, our results suggest that template lattice modulation likely plays a significant role in influencing the Ru/Ir growth behavior.

The e-Ni₃S₄@RuIr and Ni₃S₄@RuIr were loaded onto carbon support (Vulcan XC-72R) (Supplementary Fig. 9) to prevent nanoparticle aggregation, and the resulting composites (e-Ni₃S₄@RuIr/C and Ni₃S₄@RuIr/C, respectively) were annealed at 400 °C in 40% O₂ balanced with N₂ for 2 h. The transmission electron microscopy (TEM) image (Fig. 3a) of thermally oxidized e-Ni₃S₄@RuIr/C ((RuIr)O₂/C) revealed patchy Ru/Ir branches with dendritic morphology on the e-Ni₃S₄ surface. The elemental mappings (Fig. 3b) and line profiles (Fig. 3c) obtained using energy-dispersive X-ray spectroscopy (EDS) exhibited a relatively uniform distribution of Ru and Ir in the alloy form within (RuIr)O₂/C, in line with the atomic distribution observed in e-Ni₃S₄@RuIr/C prior to thermal oxidation. The lattice spacings of 0.259, 0.258, and 0.320 nm, corresponding to rutile-type oxide (01Ī), (10Ī), and (Ī10) facets, respectively, were observed in the outer regions of (RuIr)O₂/C by HAADF-STEM image and corresponding FFT patterns (parts i and ii of Fig. 3d and Supplementary Fig. 10b). In contrast, the inner region of (RuIr)O₂/C showed the (01Ī) (part iii of Fig. 3d) and (004) (Supplementary Fig. 10c) facets indicative of a residual metallic phase. The PXRD pattern of (RuIr)O₂/C (Fig. 3e) closely matched those of simulated (RuIr)O₂ (red) and residual metallic (gray box) structures, consistent with the HAADF-STEM (Fig. 3d) and high-resolution TEM (HRTEM) results (Supplementary Fig. 10).

In contrast, the thermal oxidation of Ni₃S₄@RuIr/C resulted in different Ru/Ir configurations (RuO₂@IrO₂/C; Supplementary Figs. 11, 12). RuO₂@IrO₂/C exhibited a gradient atomic distribution, with the inner and outer regions predominantly composed of Ru- and Ir-rich oxides, respectively. The local electronic structures of (RuIr)O₂/C and RuO₂@IrO₂/C were further investigated using XPS analysis (Supplementary Figs. 13, 14, and Supplementary Tables 3, 4). Both (RuIr)O₂/C and RuO₂@IrO₂/C were primarily composed of metallic species before thermal oxidation (Supplementary Fig. 13). Thermal oxidation resulted in the oxidation of the Ru phase, whereas the Ir phase maintained its metallic character and was only minimally oxidized in both (RuIr)O₂/C and RuO₂@IrO₂/C (Supplementary Fig. 14). In all cases, the Ni₃S₄-based templates were partially oxidized to NiO and partially retained as Ni₃S₄ during the thermal treatment (Supplementary Fig. 15 and Supplementary Note 3).

In summary, by regulating the lattice parameters of the Ni₃S₄ template surface, we prepared two types of Ru/Ir oxides with different atomic configurations while preserving the initial Ru/Ir atomic arrangements.

## Electrocatalytic OER performances of (RuIr)O₂/C electrocatalyst

The electrocatalytic OER performances of (RuIr)O₂/C and RuO₂@IrO₂/C were evaluated in N₂-saturated 0.1 M HClO₄ and compared with that of home-made RuO₂ nanoparticles/C (RuO₂ NPs/C) (Supplementary Fig. 16), commercial RuO₂/C (com. RuO₂/C), and commercial IrO₂/C (com. IrO₂/C). The cyclic voltammetry (CV) curves exhibited discernible reversible peaks corresponding to changes in the oxidation states of Ru and Ir within (RuIr)O₂/C and RuO₂@IrO₂/C. As shown in Supplementary Fig. 17a, two distinct redox peaks for Ru³⁺/Ru⁴⁺ (0.7 $V_{RHE}$) and Ir⁴⁺/Ir⁵⁺ (1.2–1.5 $V_{RHE}$) were observed in (RuIr)O₂/C. However, in the case of

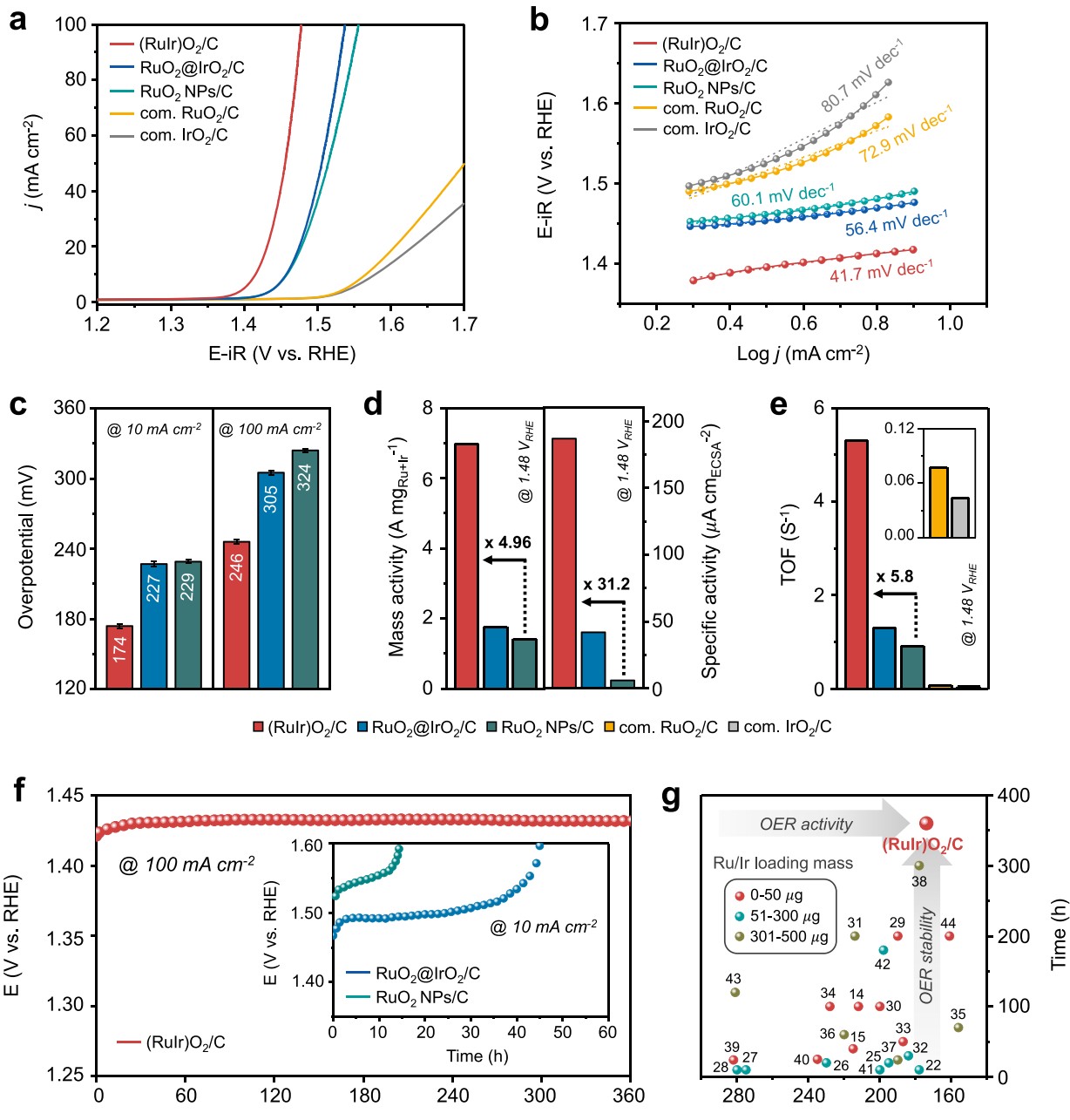

**Fig. 4 | Electrochemical OER performance. a** LSV curves of (RuIr)O$_2$/C, RuO$_2$@IrO$_2$/C, RuO$_2$ NPs/C, com. RuO$_2$/C, and com. IrO$_2$/C in 0.1 M HClO$_4$ (pH = 1.02) at a scan rate of 5 mV s$^{-1}$ and 1600 rpm. The noble metal loading was 50 µg$_{Ru+Ir}$ cm$^{-2}$ for each electrocatalysts. The measured potentials were 100% iR-compensated using the determined R$_S$ value of 12 ± 0.3 Ω. **b** Tafel plots constructed based on the curves in Fig. 4a with fitted lines for Tafel slope. **c** Overpotentials required to achieve current densities of 10 mA cm$^{-2}$ and 100 mA cm$^{-2}$, **d** mass and specific activities, and **e** TOF values at 1.48 V$_{RHE}$. **f** Chronopotentiometry test of (RuIr)O$_2$/C at 100 mA cm$^{-2}$ in 0.1 M HClO$_4$ (inset: chronopotentiometry test of RuO$_2$@IrO$_2$/C and RuO$_2$ NPs/C at 10 mA cm$^{-2}$). **g** Comparison of catalytic performances of recently reported representative Ru/Ir-based electrocatalysts with different noble metal loadings for the OER in acidic electrolytes.

RuO$_2$@IrO$_2$/C (Supplementary Fig. 17b), a redox peak corresponding to Ru$^{4+}$/Ru$^{6+}$ was observed around 0.9 V$_{RHE}$, which is not seen in (RuIr)O$_2$/C[21,22]. These results imply that the Ir in (RuIr)O$_2$/C undergoes over-oxidation instead of Ru, with rutile Ru$^{4+}$ remaining stable, whereas, in RuO$_2$@IrO$_2$/C, overoxidized Ru is present but unstable.

The linear sweep voltammetry (LSV) curves indicated that compared to the control-group catalysts, (RuIr)O$_2$/C required a lower overpotential of 174 ± 1.8 mV and 246 ± 1.2 mV to achieve a current density of 10 mA cm$^{-2}$ and 100 mA cm$^{-2}$, respectively (Fig. 4a, c). LSV curves without iR-compensation were presented in Supplementary

Fig. 18. Besides, the Tafel slope of (RuIr)O$_2$/C (41.7 mV dec$^{-1}$) was notably smaller than those of RuO$_2$@IrO$_2$/C (56.4 mV dec$^{-1}$) and RuO$_2$ NPs/C (60.1 mV dec$^{-1}$) (Fig. 4b). These results suggested that (RuIr)O$_2$/C exhibited notable catalytic activity and fast kinetics, indicating that the thorough mixing of Ru and Ir atoms considerably enhanced the OER performance. The results of electrochemical impedance spectroscopy (EIS) analysis at 1.40 V$_{RHE}$ (Supplementary Fig. 19) revealed that (RuIr) O$_2$/C showed low charge-transfer resistance and rapid electron transfer from the electrode to the catalyst surface during the OER, which agreed with the results of Tafel slope analysis.

To investigate the intrinsic activities of the Ru/Ir oxide–based catalysts, we compared their mass activities (MAs) and specific activities (SAs) at an overpotential of 250 mV (1.48 $V_{RHE}$). The active metal (Ru and Ir) loadings of the electrodes were determined by inductively coupled plasma-atomic emission spectroscopy (ICP-AES) (Supplementary Table 5). The MA of $(RuIr)O_2/C$ (6.98 A $mg_{Ru+Ir}^{-1}$) exceeded those of $RuO_2@IrO_2/C$ (1.75 A $mg_{Ru+Ir}^{-1}$) and $RuO_2$ NPs/C (1.41 A $mg_{Ru}^{-1}$) 3.98- and 4.96-fold, respectively (Fig. 4d, left). The SAs were calculated by normalizing the OER polarization curves with respect to the electrochemically active surface area (ECSA), which, in turn, was determined from the dependence of double-layer capacitance ($C_{dl}$) on different scan rate (Supplementary Fig. 20)[8,23,24]. The SA of $(RuIr)O_2/C$ (187 A $cm_{ECSA}^{-2}$) exceeded those of other samples, indicative of its comparable intrinsic activity (Fig. 4d, right and Supplementary Fig. 21). Furthermore, we calculated the turnover frequency (TOF) to demonstrate the efficiency of electrocatalysts for oxygen evolution. The TOF of $(RuIr)O_2/C$ (5.30 $s^{-1}$) surpassed those of $RuO_2@IrO_2/C$ (1.31 $s^{-1}$), $RuO_2$ NPs/C (0.92 $s^{-1}$), com. $RuO_2/C$ (0.08 $s^{-1}$), and com. $IrO_2/C$ (0.04 $s^{-1}$) (Fig. 4e).

Additionally, we synthesized unsupported $(RuIr)O_2$ and $RuO_2@IrO_2$ electrocatalysts using $SiO_2$ as a sacrificial substrate instead of carbon. This approach eliminates the potential issue of carbon corrosion during the OER operation and allows for the evaluation of Ru/Ir atomic configurations and their impact on OER performance (Supplementary Fig. 22). As shown in Supplementary Fig. 23 and Supplementary Note 4, the unsupported catalysts retained the same atomic configuration as those synthesized on carbon supports. Although the overall performance of unsupported $(RuIr)O_2$ showed a slight decrease compared to its carbon-supported counterpart ($(RuIr)O_2/C$), it still exhibited enhanced OER activity compared to unsupported $RuO_2@IrO_2$ (Supplementary Fig 24). While the carbon support enhances electrochemical performance by preventing nanoparticle aggregation and improving electrical conductivity, the Ru-Ir atomic interaction within the mixed rutile-type oxide phase remains the primary factor driving the high OER performance.

The stability of $(RuIr)O_2/C$ during the OER operation was probed by chronopotentiometry (CP) measurements at a constant current density (Fig. 4f). $(RuIr)O_2/C$ demonstrated outstanding stability, showing only a slight increase in overpotential after 360 h at 100 mA $cm^{-2}$, whereas both $RuO_2@IrO_2/C$ and $RuO_2$ NPs/C lost their initial activity within 40 and 15 h, respectively, at 10 mA $cm^{-2}$. These results underscored the inherent robustness of $(RuIr)O_2/C$ during the long-term electrochemical OER in harsh acidic environments. Notably, the OER performance of $(RuIr)O_2/C$ exceeded those of recently reported Ru- and Ir-based OER catalysts (Fig. 4g and Supplementary Table 6)[14,15,22,25–44]. To uncover the structural changes occurring during the OER, we characterized the structures of the catalysts after OER operation. The results of TEM and EDS elemental mapping analyses (Supplementary Figs. 25–27) showed that the dendritic shell thickness of $(RuIr)O_2/C$ remained unchanged, and no noticeable detachment of the Ru/Ir shell was observed, which suggested that the structural integrity of $(RuIr)O_2/C$ was preserved. In contrast, the dendritic shell of $RuO_2@IrO_2/C$ thinned due to the dendrite detachment. In addition, HRTEM image and corresponding FFT patterns (Supplementary Fig. 28) indicated that the crystallinity of $(RuIr)O_2/C$ was preserved, whereas that of $RuO_2@IrO_2/C$ deteriorated because of Ru and Ir leaching. The results of inductively coupled plasma-mass spectrometry (ICP-MS) analysis confirmed negligible leaching of Ru and Ir from $(RuIr)O_2/C$ and considerable loss of Ru and Ir from $RuO_2@IrO_2/C$ and $RuO_2$ NPs/C during long-term OER operation (Supplementary Fig. 29).

The valence electronic structures of $(RuIr)O_2/C$ and $RuO_2@IrO_2/C$ after the OER operation were examined by XPS. Compared with those of $RuO_2@IrO_2/C$, the Ru $3p_{3/2}$ (Supplementary Fig. 30) and Ir $4f$ (Supplementary Fig. 31) peaks of $(RuIr)O_2/C$ were shifted toward lower

and higher binding energies, respectively. This shift indicated a higher electron density at the Ru sites of $(RuIr)_2/C$ and implied facilitated electron transfer from Ir to Ru. Thus, in $(RuIr)O_2/C$, Ru was protected from dissolution during the OER, and the rutile-type $Ru^{4+}$ species were largely preserved, whereas $RuO_2@IrO_2/C$ contained a considerable amount of overoxidized Ru in the form of $Ru^{6+}$ ($RuO_4^{2-}$) (Supplementary Fig. 30c, d). Additionally, Ir in $(RuIr)O_2/C$ appeared as $Ir^{5+}$, which suggested that Ir was overoxidized in preference to Ru (Supplementary Fig. 31c). The notable increase in the amorphous $Ir^{3+}$ content of $RuO_2@IrO_2/C$ suggested the oxidation of Ir in the shell to porous $IrO_x$ during the OER (Supplementary Fig. 31d). This oxidation exposed the $RuO_2$ core, leading to Ru overoxidation and thereby causing Ru dissolution. The O 1 s XPS spectra (Supplementary Fig. 32 and Supplementary Table 7) revealed the changes in the contribution of lattice oxygen ($O_{M-O}$), hydroxyl groups ($O_{M-OH}$), and adsorbed water ($O_{M-H2O}$) in both $(RuIr)O_2/C$ and $RuO_2@IrO_2/C$ after OER. For $(RuIr)O_2/C$, the lattice oxygen was well retained, which emphasized the robustness of the surface oxygen species and high stability of $(RuIr)O_2/C$, whereas for $RuO_2@IrO_2/C$, the lattice oxygen content decreased. Hence, the intermixed Ir atoms in the $RuO_2$ phase emerged as a crucial factor for stabilizing the $RuO_2$ matrix during the long-term OER.

In summary, the Ir atoms positioned at neighboring cation sites alongside the Ru atoms underwent sacrificial oxidation during the long-term OER, thereby safeguarding the Ru atoms against excessive oxidation and thus increasing catalyst stability.

## Performance of $(RuIr)O_2/C$ in single-cell PEMWE

To further assess the feasibility of using $(RuIr)O_2/C$ in practical PEMWE devices, we examined its single-cell performance using membrane electrode assemblies (MEAs) (Fig. 5a and Supplementary Fig. 33). The initial measurements focused on determining the catalyst loading per unit area, which directly influences single-cell OER performance. Interestingly, the catalytic activity of $(RuIr)O_2$ deteriorated with increasing catalyst loading, indicating that the MEA thickness considerably affected the charge transport efficiency (Supplementary Fig. 34a)[45,46]. This observation was further supported by EIS analysis (Supplementary Fig. 34b, c), which revealed that both ohmic and charge-transfer resistances increased with thicker MEAs, underscoring the need to optimize catalyst loading for balancing activity and efficient charge transfer within the MEA[47]. Consequently, the examination of changes in the catalytic activity of $(RuIr)O_2/C$ showed that charge transfer was facilitated even at a low catalyst loading (0.25 mg $cm^{-2}$), which resulted in high catalytic activity.

Next, we compared the activity of commercial $IrO_2$ (com. $IrO_2$) in a system using the same single-cell material. At a cell voltage of 2.0 V, $(RuIr)O_2/C$ achieved a high current density of 4.96 A $cm^{-2}$, which was 56% higher than that observed for com. $IrO_2$ (3.17 A $cm^{-2}$) (Fig. 5b) and exhibited a notable MA of 19.84 A $mg_{Ru+Ir}^{-1}$ at a loading of only 0.25 $mg_{Ru+Ir}$ $cm^{-2}$ (Fig. 5c). The single-cell performance of $(RuIr)O_2/C$ demonstrates enhanced efficiency, achieving high activity even at low catalyst loading, highlighting its intrinsic capability for the OER. Furthermore, the EIS results obtained for $(RuIr)O_2/C$ and com. $IrO_2/C$ in the high-voltage region (2.0 V), where the mass-transfer effects were most pronounced, revealed enhanced membrane contact and rapid charge transfer on the surface of $(RuIr)O_2/C$ (Supplementary Fig. 35). The long-term assessment of the PEMWE performance using MEAs with $(RuIr)O_2/C$ and com. $IrO_2$ at a constant current density of 1.0 A $cm^{-2}$ (Fig. 5d) showed that the $(RuIr)O_2/C$ maintained a stable cell voltage over 250 h with a very low deterioration rate of 0.08 mV $h^{-1}$. Additionally, SEM analysis was conducted to assess the surface morphology and cross-section of the MEAs before and after the single-cell durability test, specifically examining catalyst detachment and dissolution behavior during PEWWE operation. Notably, the commercial $IrO_2$ (with a reduction rate of 40.37%) exhibited a significant decrease in catalyst layer thickness, whereas the $(RuIr)O_2/C$ showed only a

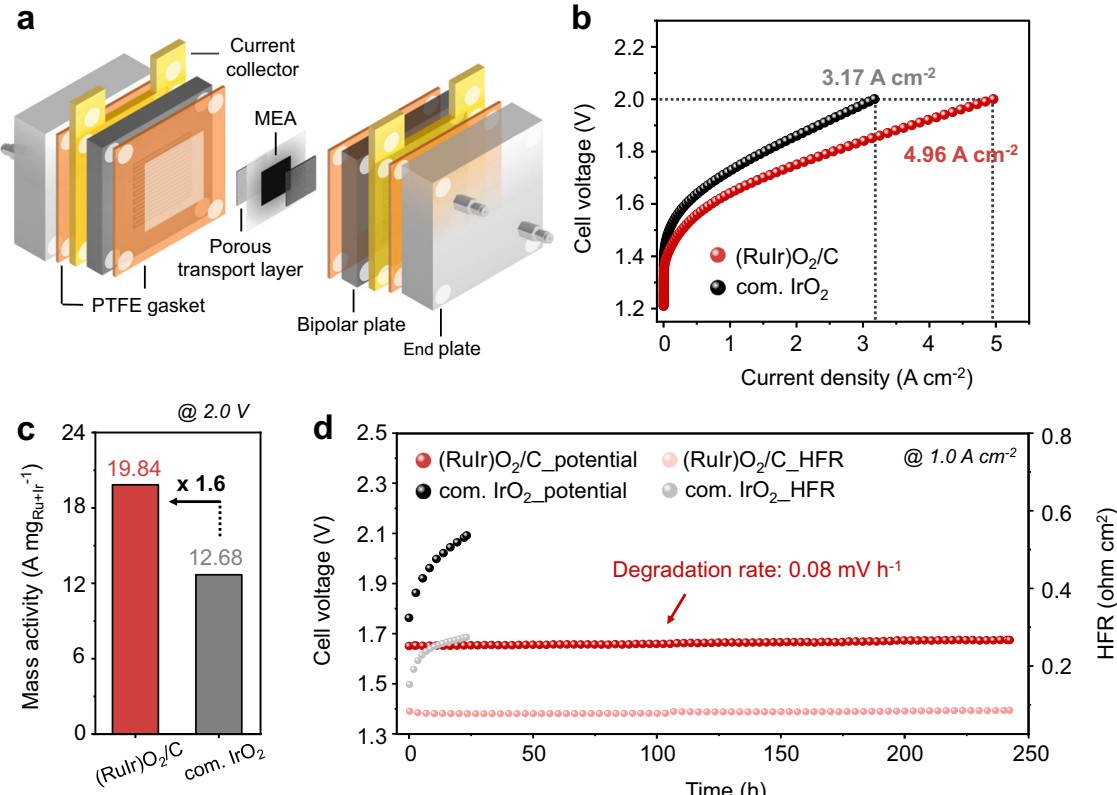

**Fig. 5 | Single-cell performances in PEMWE. a** Schematic illustration of the membrane-electrode assembly used for PEMWE. **b** Polarization curves for PEMWE using (RuIr)O$_2$/C and com. IrO$_2$ as anode catalysts and commercial Pt/C as cathode catalyst in 0.1 M HClO$_4$ at 80 °C. The noble metal loadings were 0.25 mg$_{Ru+Ir}$ cm$^{-2}$ for the anode and 1 mg$_{Pt}$ cm$^{-2}$ for the cathode. Current densities at a cell voltage of 2.0 V are 4.96 A cm$^{-2}$ and 3.17 A cm$^{-2}$ for (RuIr)O$_2$/C and com. IrO$_2$, respectively. No cell volatges were iR compensated. **c** Mass activities (A mg$_{Ru+Ir}^{-1}$) of (RuIr)O2/C and com. IrO$_2$ at a cell voltage of 2.0 V. **d** OER stability test of PEMWE cells with (RuIr)O$_2$/C and com. IrO$_2$ recorded at 1.0 A cm$^{-2}$. The HFR on the right y-axis means high-frequency resistance.

modest reduction of about 28.66% during the durability test (Supplementary Figs. 36, 37). This demonstrates that the (RuIr)O$_2$/C remained robust with minimal degradation, effectively preserving its OER performance over prolonged PEMWE operation.

**Origin of the improved OER activity and stability of (RuIr)O$_2$/C**
To elucidate the origin of the improved OER activity and stability of (RuIr)O$_2$/C, we examined changes in the chemical states of Ru and Ir during the OER using in situ Ru K-edge and Ir L$_3$-edge X-ray absorption fine structure (XAFS) spectroscopy at applied potentials of 0.4–1.6 V$_{RHE}$ (Supplementary Fig. 38). The Ru K-edge X-ray absorption near-edge structure (XANES) spectra (Fig. 6a) showed a minimal positive shift in the pre-edge position with increasing potential. The Ru K-edge position, determined from the first derivatives of the XANES spectra (Supplementary Fig. 39), was plotted as a function of the Ru oxidation state (Fig. 6b). Ru foil (Ru$^0$), Ru(acac)$_3$ (Ru$^{3+}$; acac = acetylacetonate), and RuO$_2$ (Ru$^{4+}$) powders were used as reference materials, and a slope of 2.753 eV per oxidation-state unit was obtained. The Ru oxidation state increased from 3.69 at +0.4 V$_{RHE}$ to 4.11 at +1.2 V$_{RHE}$, while the change in the valence state at higher potentials was negligible. Therefore, the Ru$^{4+}$ states in (RuIr)O$_2$/C remained securely intact at a high overpotential (+1.6 V$_{RHE}$), effectively preventing the overoxidation of Ru species during the OER.

The changes in the Ir oxidation state were very different from those in the Ru oxidation state. The Ir L$_3$-edge XANES spectrum of (RuIr)O$_2$/C (Fig. 6c) showed broad white lines (WLs) corresponding to the 2p-to-5d transition[48,49]. In addition, Ir L$_3$-edge XANES analysis revealed a pronounced positive shift in the WL position and intensity upon a potential increase from 0.4 to 1.6 V$_{RHE}$. The quantitative change

in the Ir d-band states was analyzed using d-band hole counts derived from the WL positions, which were determined from the second derivatives of the XANES spectra (Supplementary Fig. 40)[49–51]. The Ir L$_3$-edge WL position as a function of the formal d-band hole count was depicted in Fig. 6d. A slope of 0.812 eV per d-band hole was obtained for Ir black (5d$^7$6s$^2$), Ir(acac)$_3$ (5d$^6$6s$^0$), and IrO$_2$ (5d$^5$6s$^0$)[50–52]. A notable depopulation of Ir d-band states with increasing potential was observed, with Ir d-band hole counts of 3.18, 5.14, 5.64, and 5.66 $e^-$ observed at +0.4, +1.2, +1.45, and +1.6 V$_{RHE}$, respectively. In summary, (RuIr)O$_2$/C showed a considerable depopulation of Ir d-band states with increasing potential (Δ$d$ = 2.48 for 0.4–1.6 V$_{RHE}$), whereas the Ru oxidation state was maintained at approximately +4 in the 1.2–1.6 V$_{RHE}$ range (Fig. 6e). Therefore, Ir with a substantial number of d-band holes was concluded to boost OER activity while suppressing the detrimental overoxidation of Ru within the mixed rutile-type phase.

Alterations in the local coordination structures around the Ru and Ir atoms in (RuIr)O$_2$/C were further investigated using in situ extended X-ray absorption fine structure (EXAFS) spectroscopy. The Ru K-edge and Ir L$_3$-edge Fourier-transformed (FT) EXAFS spectra were subjected to least-squares fitting, with the thus derived structural parameters summarized in Supplementary Tables 8, 9. Fig. 7a presents Ru−O and Ir−O bond lengths as functions of the applied potential. The Ir−O bond length of (RuIr)O$_2$/C decreased from 1.994 to 1.966 Å upon a potential increase from +1.2 to +1.6 V$_{RHE}$ (Fig. 7b), which indicated the considerable oxidation of Ir during the OER not attributable to hole-doped states induced by surrounding vacancy formation (Fig. 7d)[51]. Conversely, Ru−O bond length exhibited a different behavior, being considerably larger at +1.45 V$_{RHE}$ (2.025 Å) than at +1.2 V$_{RHE}$ (1.989 Å) and +1.6 V$_{RHE}$ (1.963 Å) (Fig. 7c). This Ru−O bond elongation during the

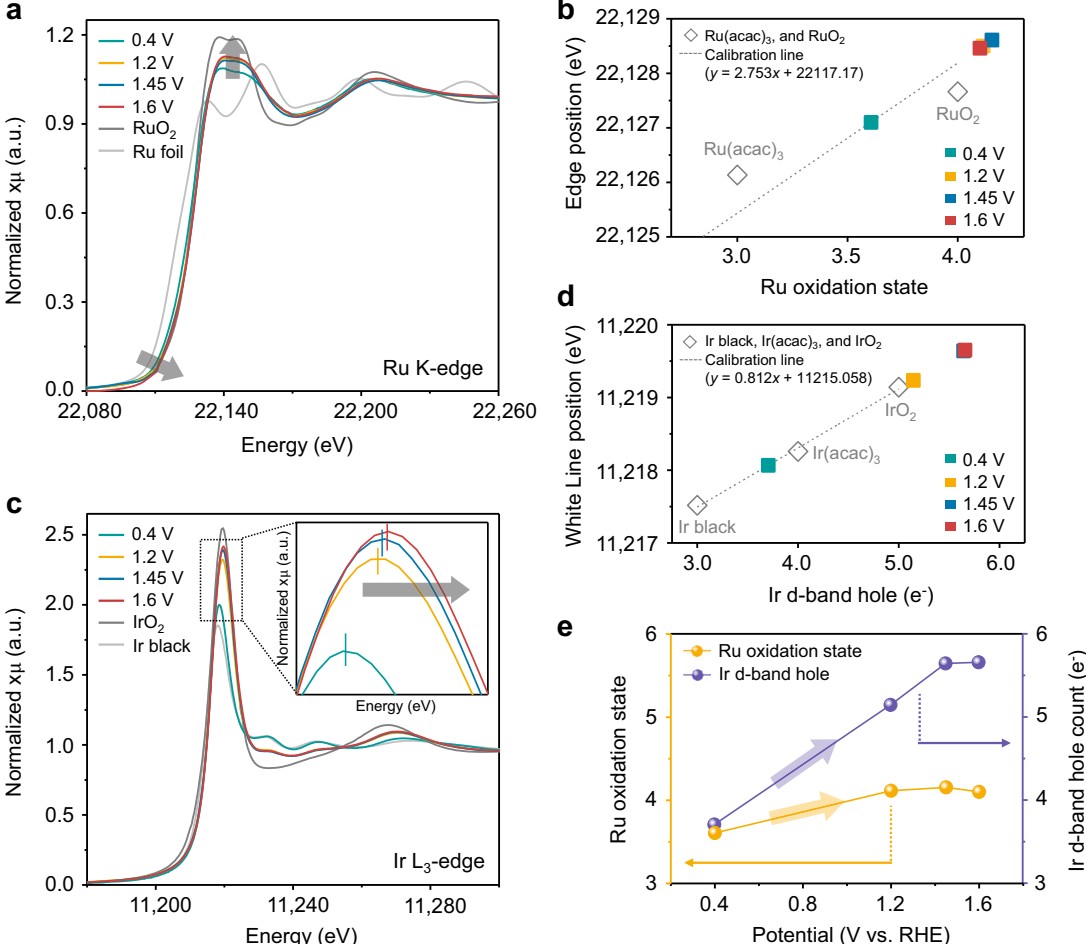

**Fig. 6 | In-situ XANES analysis. a** Ru K-edge XANES spectra of (RuIr)O₂/C at different applied potentials in O₂-saturated 0.1 M HClO₄. **b** Change in the Ru K-edge position of (RuIr)O₂/C as a function of the Ru valence state. **c** Ir L₃-edge XANES spectra of (RuIr)O₂/C recorded at different applied potentials in O₂-saturated 0.1 M

HClO₄. **d** Change in the Ir L₃-edge white line position of (RuIr)O₂/C as a function of the formal d-band hole count. **e** Change in the Ru oxidation state (yellow, left) and Ir formal d-band hole count (purple, right) upon an increase in applied potential from 0.4 to 1.6 $V_{RHE}$.

on-site catalytic stage (from +1.2 $V_{RHE}$ to +1.45 $V_{RHE}$) could weaken the electron-withdrawing effect of the neighboring coordinated oxygen atoms and thus decrease the Ru oxidation state to prevent catalyst dissolution[53]. Furthermore, by examining the M–O distance as a function of the Ir d-band hole count (Fig. 7d) and Ru oxidation state (Fig. 7e), we confirmed that the (RuIr)O₂/C-catalyzed OER proceeded via the AEM pathway, i.e., did not involve lattice oxygen, which resulted in high catalyst stability.

To validate the experimentally determined alteration in the M–O bond length with changes in the OER potential, we analyzed the variation in the Ru–O and Ir–O bond lengths on the catalyst surface during deprotonation using DFT calculations. Previous studies have highlighted the surface transition from OH* termination to O* termination in rutile-type structures under an applied oxidation potential[54]. In this context, we constructed surface structures with different numbers of adsorbed H* species and examined the lengths of surface metal–oxygen bonds (Fig. 7f, g). Interestingly, the Ir–O bond length decreased upon surface deprotonation, whereas certain Ru–O bond lengths concomitantly increased. This behavior suggested that the Ru–O bond elongation in (RuIr)O₂ at +1.45 $V_{RHE}$, as observed by in situ EXAFS spectroscopy, was due to the elongation of the Ru–O bond adjacent to the deprotonated $O_{2c}$ (Supplementary Fig. 41). Additionally, as deprotonation progressed up to +1.6 $V_{RHE}$, contraction of the

Ru–O bond was observed. Theoretical analysis suggested that this is due to the deprotonation of $HO_{2c}$ bonded to Ru at oxidative potentials, resulting in the formation of $O_{2c}$. To further explore the effect of metal type on M–O bond length changes, we analyzed the M–O bond lengths of pure rutile-type oxides (IrO₂, RuO₂) and $(Ru_{AS}Ir_{M6c})O_2$, where Ir and Ru were located at $M_{6c}$ and AS, respectively (Supplementary Fig. 42). Upon surface deprotonation, the $X_{AS}$–O (X = Ir, Ru) bond length decreased in all cases, whereas some $X_{M6c}$–O bond lengths increased. The above findings suggest that the positioning of Ru at $M_{6c}$ within the rutile-type (RuIr)O₂ increased the Ru–O bond length, which was consistent with in situ EXAFS results.

### Enhanced OER mechanism of (RuIr)O₂/C via AEM pathway

To theoretically rationalize the notable OER performance of the (RuIr)O₂ catalyst compared with those of RuO₂ and IrO₂, we generated (110) surfaces for RuO₂, IrO₂, and (RuIr)O₂ using their bulk lattice parameters (Supplementary Note 5). Given the considerable thickness of the outer shell of RuO₂@IrO₂, its computational modeling was similar to that of IrO₂. The surface structure of (RuIr)O₂ was constructed by considering the oxidation state of Ir and its position within the surface structure (Supplementary Figs. 43–46 and Supplementary Note 6). Various Ir positions, including active sites (AS), six-coordinated metal sites ($M_{6c}$), subpositions of active sites (sub-AS), and subpositions of

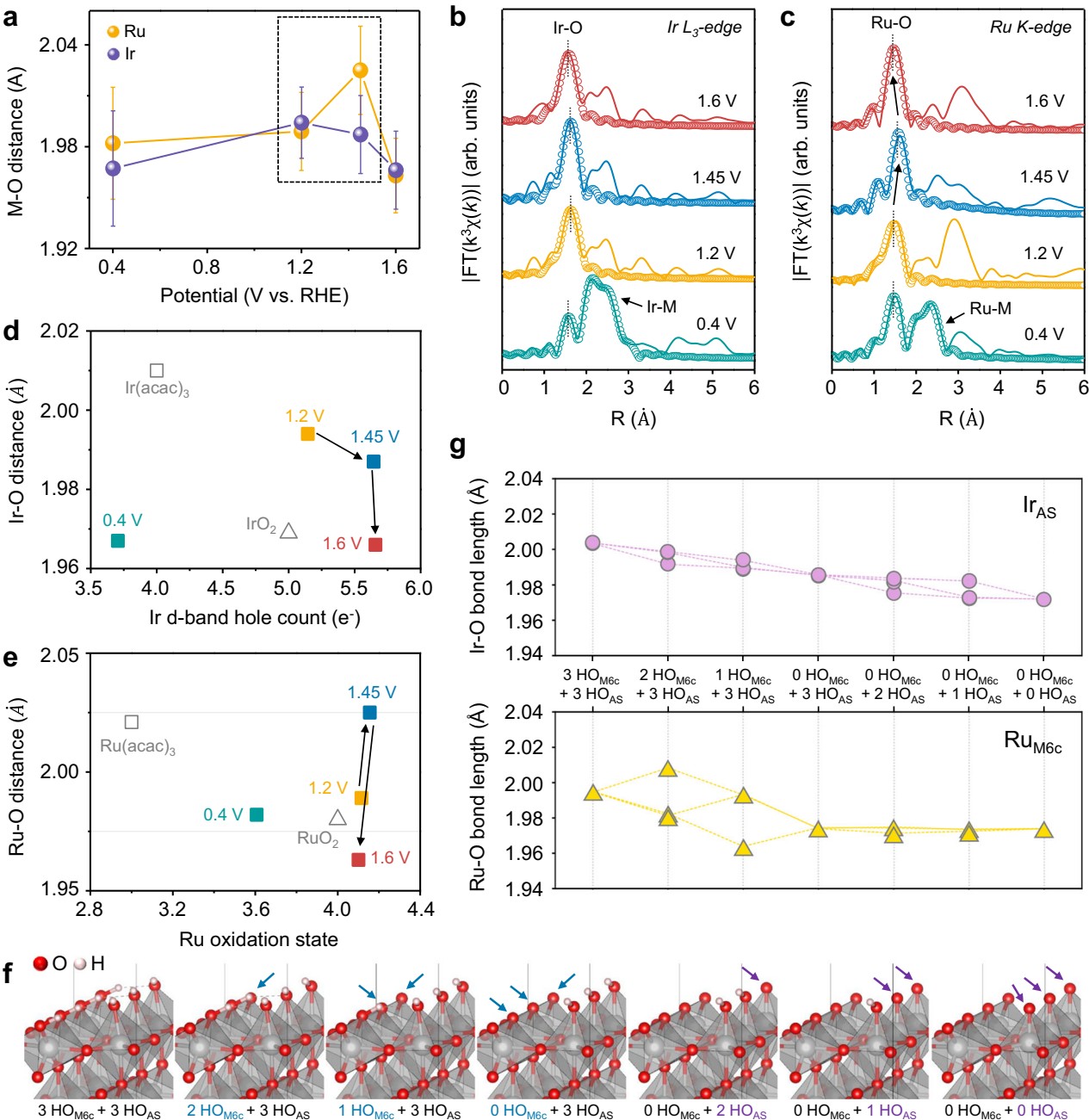

**Fig. 7 | In-situ EXAFS analysis and DFT calculations for M–O bonding. a** Ru–O (red) and Ir–O (blue) bond lengths in (RuIr)O₂/C obtained by the analysis of in situ FT-EXAFS spectra. In situ **b** Ir L₃-edge and **c** Ru-K edge FT-EXAFS spectra of (RuIr)O₂/C obtained at different applied potentials in O₂-saturated 0.1 M HClO₄. Experimental operando **d** Ir–O and **e** Ru–O distances of (RuIr)O₂/C as functions of Ir d-band hole count and Ru oxidation state, respectively, under applied potentials of +0.4, +1.2, +1.45, and +1.6 V$_{RHE}$. **f** Atomic configurations with different coverages calculated for H-covered rutile (110) surface structures (HO$_{M6c}$ and HO$_{AS}$ are H atoms bonded to O atoms on six-coordinated metal sites (M$_{6c}$) and active sites (AS), respectively.) **g** DFT-calculated M–O bond lengths (Boltzmann-averaged) for all (RuIr)O₂ surface structures according to surface coverage. Ir$_{AS}$ and Ru$_{M6c}$ correspond to Ir at AS and Ru at M$_{6c}$, respectively.

six-coordinated metal sites (sub-M$_{6c}$), were examined. Based on the results of in situ XANES analysis (Fig. 6), all Ir atoms were placed at the AS, where they exhibited the highest oxidation states. The cations were then arranged to achieve a Ru/Ir atomic ratio of 1:1 in each layer. In this manner, we created all possible (RuIr)O₂ surface structures (total: $_6C_3 × _6C_3 = 400$), from which 38 unique surface structures were extracted (Fig. 8a and Supplementary Figs. 47, 48). Utilizing these surface structures, we calculated the Gibbs free energy changes of OER intermediates for all sites across 38 structures and 3 sites

(114 calculations in total), following the AEM. The corresponding weighted averages were then determined using the Boltzmann probability, with more stable surface structures contributing more significantly to material properties (Supplementary Fig. 49). During the AEM pathway (Fig. 8b, c), the Ir active sites in (RuIr)O₂ exhibited weaker affinities for all oxygen intermediates (OH*, O*, OOH*) compared to those in IrO₂. Consequently, (RuIr)O₂ demonstrated a higher OER activity (overpotential ($\eta$) = 0.48 V) than pure rutile-type oxides ($\eta$ = 0.76 V for IrO₂, $\eta$ = 0.62 V for RuO₂). Although (RuIr)O₂ exhibited

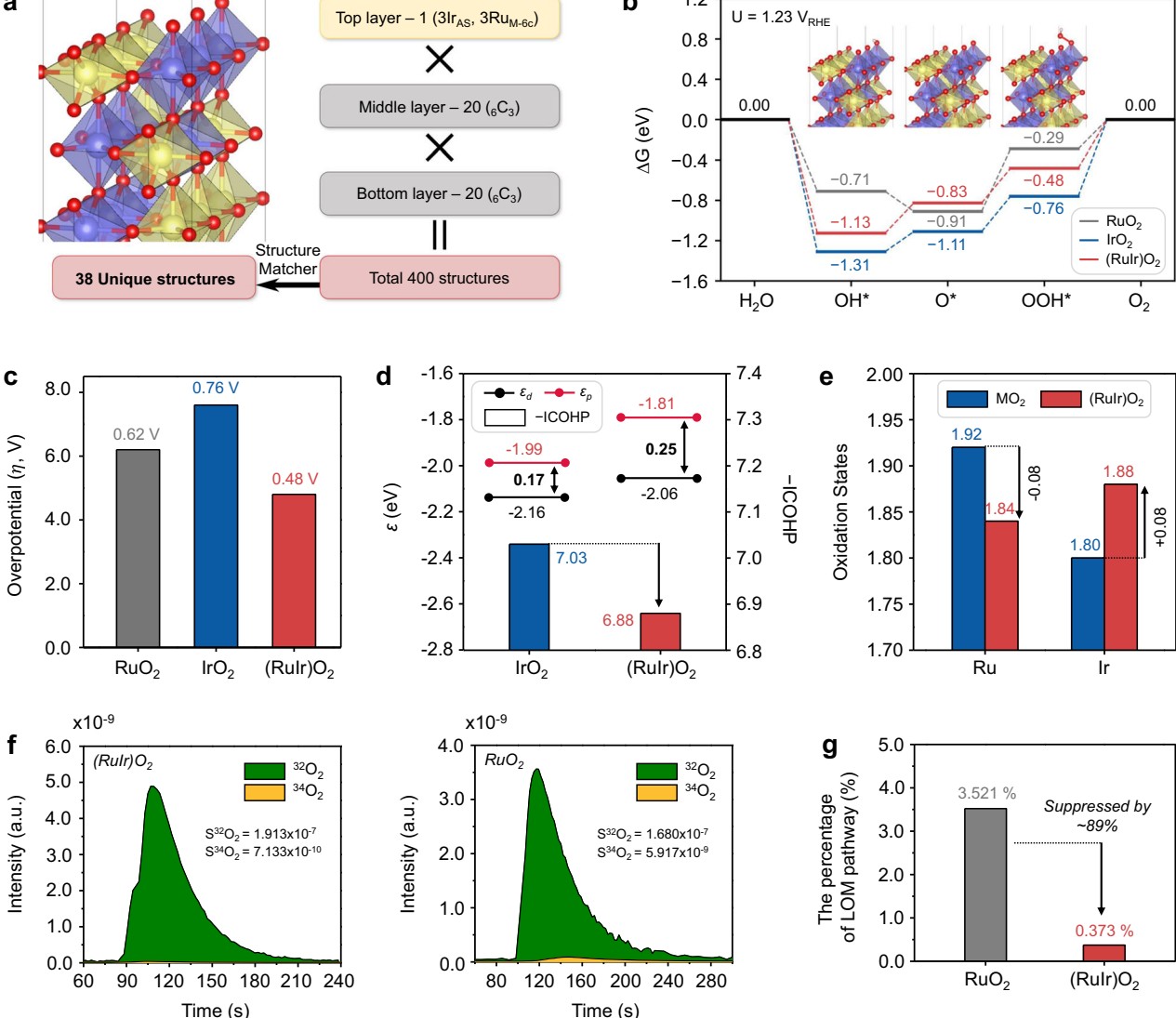

**Fig. 8 | DFT calculations and DEMS analysis for OER mechanism. a** Schematic illustration of (RuIr)O$_2$ structure generation. Red, yellow, and blue spheres correspond to O, Ru, and Ir atoms, respectively. **b** Gibbs free energy diagram for the adsorption of OER intermediates on RuO$_2$, IrO$_2$, and (RuIr)O$_2$ at 1.23 V$_{RHE}$. **c** Theoretical OER overpotentials of RuO$_2$, IrO$_2$, and (RuIr)O$_2$. **d** $d$-band center ($\varepsilon_d$) and $p$-band center ($\varepsilon_p$) positions for surface Ir and adsorbed O*, respectively, and Boltzmann-averaged −ICOHP values of Ir−O* bonding for IrO$_2$ and (RuIr)O$_2$. Smaller −ICOHP values indicate weaker M−O bonding. **e** Average oxidation states of surface atoms for MO$_2$ (M = Ru, Ir) and (RuIr)O$_2$. **f** DEMS signals of $^{34}$O$_2$ ($^{16}$O$^{18}$O) and $^{32}$O$_2$ ($^{16}$O$^{16}$O) from the evolved O$_2$ for the $^{18}$O-labeled (RuIr)O$_2$/C (left) and homemade-RuO$_2$/C (right) in 0.1 M HClO$_4$ solution in H$_2$$^{16}$O. **g** Percentage contribution of lattice oxygen (LOM%) in the OER.

better OER activity than RuO$_2$ and IrO$_2$ through the LOM pathway (Supplementary Fig. 50 and Supplementary Note 7), the calculated overpotentials were significantly higher compared to those of the AEM pathway. Therefore, (RuIr)O$_2$ exhibits notable OER activity compared to RuO$_2$ and IrO$_2$, particularly highlighting the efficiency of the OER process via the AEM pathway.

The difference in energy levels ($|\varepsilon_d - \varepsilon_p|$) between the $d$-band center ($\varepsilon_d$) of the active-site metal and $p$-band center ($\varepsilon_p$) of adsorbed O* reflects the M−O interaction strength[55]. Specifically, chemical binding properties are determined by electron transfer between the metal d-orbital and adsorbed O* p-orbital, with smaller differences between $\varepsilon_d$ and $\varepsilon_p$ indicating stronger M−O bonding[56]. In this regard, we observed an increase in $|\varepsilon_d - \varepsilon_p|$ from 0.17 (IrO$_2$) to 0.25 ((RuIr)O$_2$), which suggested weakened M−O bonding between the O* adsorbate and Ir active sites in (RuIr)O$_2$ (Fig. 8d and Supplementary Fig. 51). Additionally, the − integrated crystal orbital Hamilton population (−ICOHP) for M−O*, which indicates bond strength, was lower for (RuIr)O$_2$ than for IrO$_2$ (Fig. 8d and Supplementary Fig. 52). These

results imply that the presence of both Ru and Ir at the cation sites of the rutile-type oxide weakened metal−adsorbate bonding, thereby increasing the OER activity of (RuIr)O$_2$. Furthermore, we examined the oxidation states of the surface cations in the rutile-type oxides (Fig. 8e). Bader charge analysis revealed a decrease (increase) of 0.08 in the oxidation state of Ru (Ir) in the mixed rutile-type oxide, indicating electron transfer from Ir to Ru. This finding suggests that overoxidation inhibited Ru dissolution and thus increased catalyst stability under harsh OER conditions[9,23,57]. Overall, (RuIr)O$_2$/C demonstrated enhanced OER performance attributed to the synergistic effect of increased Ir activity and Ru stability.

We further carried out in-situ differential electrochemical mass spectrometry (DEMS) analyses using the isotope $^{18}$O to investigate to verify the suppressed lattice oxygen participation on (RuIr)O$_2$ catalysts during the OER[58,59]. Before DEMS measurement, the $^{18}$O-labeled (RuIr)O$_2$ and home-made RuO$_2$ catalysts were prepared by CV cycling in a 0.1 M HClO$_4$ in heavy-oxygen water (H$_2$$^{18}$O). Then, the evolved O$_2$ was measured by DEMS in a 0.1 M HClO$_4$ electrolyte of H$_2$$^{16}$O

(Supplementary Fig. 53). The signals of the $^{34}O_2$ indicate the direct $^{16}O$-$^{18}O$ coupling, where the $^{16}O$ originates from water and $^{18}O$ originates from the lattice oxygen[60]. The participation ratio of lattice oxygen (LOM%) was evaluated by the ratio of $^{34}O_2$ to $(^{32}O_2 + {}^{34}O_2)$. As shown in Fig. 8f, g, the LOM% of the $(RuIr)O_2$ was only 0.373%, whereas the LOM% of the homemade-$RuO_2$ (3.521%) was about ~9.4-fold higher than that of $(RuIr)O_2$. Therefore, the lattice oxygen participation during the OER was significantly hindered in the $(RuIr)O_2$, which corroborates with its high OER stability over $RuO_2$ under acidic conditions.

## Discussion

In summary, efficient rutile-type $(RuIr)O_2/C$ electrocatalysts with notable intrinsic activity and stability at 100 mA cm$^{-2}$ of current density were developed through the changeable growth behavior of Ru/Ir atoms on lattice-parameter-modulated templates. The results of in situ XAS analysis and DFT calculations demonstrated that the improved catalytic performance arises from the maximized synergy between the Ru and Ir atoms mixed at the atomic level. The Ir atoms stabilized the local coordination environment around Ru, promoting the AEM and adjusting the valence electronic structure of the Ru sites to prevent Ru overoxidation and achieve a durable OER during prolonged operation. Moreover, the atomic-scale coexistence of Ru and Ir resulted in optimized oxygen-intermediate adsorption energy and outstanding catalytic activity. We expect that the modulation of the surface parameters of template can be leveraged to form various alloy materials and develop new electrocatalytic materials.

## Methods

### Chemicals

$NiCl_2 \cdot H_2O$ (99.95% trace metal basis), $Ir(acac)_3$ (97%), $Ru(acac)_3$ (97%), 1,2-hexadecanediol (technical grade, 90%), 1-dodecanethiol (98%), oleic acid (technical grade, 90%), oleylamine (OAm, technical grade, 70%), and 1-octadecene (technical grade, 90%) were purchased from Sigma-Aldrich. All chemicals were used as received without further purification.

### Synthesis of pristine $Ni_3S_4$ and e-$Ni_3S_4$ templates

The pristine $Ni_3S_4$ template was synthesized using a previously reported method with minor modifications[61]. A slurry of $NiCl_2 \cdot H_2O$ (0.2 mmol), 1,2-hexadecanediol (0.4 mmol), oleic acid (0.2 mL), 1-octadecene (5 mL), and OAm (1.2 mL) was prepared in a 100 mL Schlenk tube equipped with a magnetic stirrer. The tube was placed in an oil bath held at 90 °C, evacuated for 60 min, and charged with Ar (1 atm). After the injection of 1-dodecanethiol (0.5 mL), the tube was placed in a preheated oil bath held at 120 °C and maintained at this temperature for 40 min. Finally, the oil bath was heated to 225 °C for more than 15 min. The reaction mixture was cooled to room temperature, and the dark precipitate was washed with isopropanol/acetone (15 mL/15 mL) and collected via centrifugation. The washing process was repeated twice. The precipitate was dispersed in toluene for the further growth of Ru and Ir. To synthesize the e-$Ni_3S_4$ template, the sample was further heated from 225 to 240 °C. When the oil bath temperature reached 240 °C, a stock solution of $Ir(acac)_3$ (0.025 mmol) in OAm (3 mL) was injected into the Schlenk tube, and the reaction mixture was maintained at 240 °C for 5 h. The washing process was identical to that used for the pristine $Ni_3S_4$.

### Synthesis of $Ni_3S_4$@RuIr and e-$Ni_3S_4$@RuIr

For e-$Ni_3S_4$@RuIr synthesis, a 100 mL Schlenk tube with a stirring bar was charged with the e-$Ni_3S_4$ template (15 mg), $Ru(acac)_3$ (0.630 mmol), $Ir(acac)_3$ (0.245 mmol), and OAm (25 mL), vacuumed in an oil bath at 80 °C for 20 min, and charged with Ar (1 atm). Thereafter, the tube was transferred to a preheated oil bath and held at 240 °C for 2 h. The dark precipitate was sequentially washed with toluene (10 mL) and methanol (10 mL) and collected by centrifugation. This process was repeated twice. $Ni_3S_4$@RuIr with a Ru/Ir gradient core@shell

atomic configuration was prepared using the pristine $Ni_3S_4$ template instead of the e-$Ni_3S_4$ template.

### Synthesis of $(RuIr)O_2/C$ and $RuO_2$@$IrO_2/C$

The $(RuIr)O_2/C$ and $RuO_2$@$IrO_2/C$ were prepared using the heat treatment and oxidative conditions described for e-$Ni_3S_4$@RuIr and $Ni_3S_4$@RuIr, respectively. The as-prepared e-$Ni_3S_4$@RuIr and $Ni_3S_4$@RuIr were mixed with Vulcan carbon (XC-72R) in chloroform to achieve a catalyst loading of 20 wt%. The composites were collected by centrifugation and stored in a vacuum chamber. The dried powders were annealed at 400 °C for 2 h in a flow of $N_2$-balanced 40% $O_2$ (0.5 mL min$^{-1}$) inside a tube furnace.

### Synthesis of home-made $RuO_2$ NPs/C

The home-made $RuO_2$ nanoparticles (NPs) were synthesized using a previously reported method with minor modifications[14,16]. To prepare home-made $RuO_2$ NPs, Ru NPs were synthesized first. A slurry of $Ru(acac)_3$ (0.06 mmol), CTAC (0.01 mmol), 1,2-hexadecanediol (0.4 mmol), and OAm (10 mmol) was prepared in a 100 mL Schlenk tube equipped with a magnetic stirrer. The slurry was vacuumed in an oil bath at 30 °C for 10 min and charged with Ar (1 atm). Thereafter, the tube was transferred to a preheated oil bath and held at 280 °C for 2 h. The product was sequentially washed with ethanol (15 mL) and toluene (10 mL) and collected by centrifugation. The resulting Ru NPs were supported on carbon black and thermally annealed at 400 °C for 2 h in a flow of $N_2$-balanced 40% $O_2$ (0.5 mL min$^{-1}$) inside a tube furnace, which was identical to the experimental method described above to prepare $(RuIr)O_2/C$ and $RuO_2$@$IrO_2/C$.

### Preparation of the working electrode

To prepare the working electrode, 5 mg of the carbon-supported catalyst was mixed with 125 μL of isopropanol, 25 μL of Nafion (5 wt%, Alfa-Aesar), and 350 μL of deionized water. The mixture was sonicated in an ice bath for 30 min to obtain a well-dispersed catalyst ink.

Prior to loading the catalyst ink, the rotating disk electrode (RDE) was polished on a micro-cloth pad using 1 μm of alumnia suspension (AS-100, Alpha alumina, MOHS 9) and 1 μm of monocrystalline diamond suspension (MetaDi) solutions. Subsequently, 7 μL of the catalyst ink was drop-cast onto a glassy carbon electrode (5.0 mm diameter; disk geometric area: 0.1963 cm$^2$) to serve as the working electrode. The resulting mass loading of the catalyst on the electrode was 356.59 μg cm$^{-2}$. Considering the nanoparticles were supported on the carbon support at 20 wt%, the total metal mass loading on the electrode area was calculated to be 71 μg cm$^{-2}$, including a noble metal loading of 50 μg$_{Ru+Ir}$ cm$^{-2}$. The electrode was then vaccum-dried at room temperature before use.

For stability evluations using the choronopotentiometric method, 28 μL of the catalyst ink was drop-cast onto the electrode, resulting in a noble metal loading of 200 μg$_{Ru+Ir}$ cm$^{-2}$.

### Electrochemical characterization

Electrochemical measurements were performed at room temperature in a typical three-electrode cell using a CHI750E electrochemical analyzer (Bi-Potentiostat, CH Instruments) with $N_2$-saturated 0.1 M $HClO_4$ electrolyte. An SVC-3 voltammetry cell with teflon cap (BAS Inc.) was used, and freshly prepared 0.1 M $HClO_4$ electrolyte was utilized for each measurement. A graphite carbon rod (6 mm, Qrins) served as the counter electrode, and an Ag/AgCl electrode (saturated 3 M NaCl, ALS Co., Ltd) was used as the reference electrode.

All of the potentials in this work are reported relative to the reversible hydrogen electrode (RHE) scale. The potentials were converted using the equation:

$$E_{RHE} = E_{mea} + E_{Ag/AgCl} + 0.0591 \times pH,$$

where $E_{RHE}$ is the potential on the RHE scale, $E_{mea}$ is the meausred experimental potential, and $E_{Ag/AgCl}$ is the reference electrode potential (0.1976 $V_{RHE}$). The pH of the 0.1 M HClO4 electrolyte was measured to be 1.02 on average, based on eight measurements using a PHS-3D-02 pH meter (Shanghai San-Xin Instrumentation, Inc.). Addtionally, all polarization curves were corrected for 100% iR-compensation.

Before evaluating OER performance in the half-cell system, the electrolyte was purged with high-purity $N_2$ gas (99.999%) for 30 min. Electrochemical pretretment was conducted by performing 20 cyclic voltammetry (CV) cycles in the potential range of 0.05–1.1 $V_{RHE}$ at a scan rate of 0.2 V s$^{-1}$ to clean and stabilize the catalyst surface. OER activity was assessed using linear sweep voltammetry (LSV) in the potential range of 1.1–1.8 $V_{RHE}$ at a scan rate of 5 mV s$^{-1}$ and an electrode rotation speed of 1600 rpm.

Electrochemical impedance spectroscopy (EIS) was performed in the frequency range of 100 kHz to 0.1 Hz using a small AC perturbation with a 5 mV amplitude at a fixed potential of 1.45 $V_{RHE}$. The Nyquist plot was used to determine the solution resistance ($R_S$), identified as x-axis intercept in the high-frequency region. At high frequencies, contributions from charge transfer resistance and double-layer capacitance are negligible, leaving only the solution resistance. The measured $R_S$ was used for iR-compensation to correct for voltage lossess due to solution resistance. In our experiments, $R_S$ was determined to be 12 $\pm$ 0.3 Ω.

Stability was evalutated using chronopotentiometry at a constant current density of 100 mA cm$^{-2}$ for (RuIr)$O_2$/C and 10 mA cm$^{-2}$ for $RuO_2@IrO_2$/C and home-made $RuO_2$/C, respectively, in $N_2$-saturated 0.1 M HClO4.

The electrochemical surface area (ECSA) was determined from the double-layer capacitance ($C_{dl}$), which was calculated by recording CV curves at different scan rates (20, 40, 80, 100, 150, and 200 mV s$^{-1}$) in the non-faradaic potential region (0.30–0.48 $V_{RHE}$). The charging current ($i_c$) at the center potential was evaluated as the product of the scan rate ($v$) and $C_{dl}$:

$$i_c = vC_{dl}.$$

The ECSA was derived using the equation:

$$ECSA = C_{dl}/C_s,$$

where $C_s = 0.035$ mF cm$^{-2}$.

The turnover frequency (TOF) was calculated using the equation:

$$TOF = (J \times A \times \zeta)/(4 \times F \times n_{mass}),$$

where $J$ is the current density at 250 mV, $A$ is the geometric area of the electrode, $\zeta$ is the faradaic efficiency (assumed to equal 100%), $F$ is the Faraday constant, and $n_{mass}$ is the number of moles of active sites (Ru and Ir), determined from the ECSA. The total mass of Ru and Ir was calculated using:

$$n_{mass} = (n_{Ru+Ir} \times N_A)/(M_W)$$

where $n_{Ru+Ir}$ is the total mass of Ru and Ir loaded onto the electrode, $N_A$ is Avogadro's constant, and $M_W$ is the molecular weight.

## PEMWE single-cell test

(RuIr)$O_2$/C or comm. IrO2 (99.99%, Alfa Aesar, USA) and Pt/C (46.9% Pt, Tanaka, Japan) were used as the anode and cathode catalysts, respectively. All tests were performed with an MEA prepared using the catalyst spray-coated membrane method. The catalyst slurry was prepared by mixing the catalyst with deionized water, the Nafion ionomer, and isopropanol, followed by sonication for 30 min at 40 °C. A Nafion 212 membrane, composed of entirely of Nafion polymer and with a nominal thcickness of approximately 50 μm, was used. For single-cell testing, the catalysts were uniformly spray-coated onto a 4 cm$^2$ active area of a 25 cm$^2$ membrane using a spray machine. The loadings were (RuIr)$O_2$/C at 0.25–1 mg$_{Ru+Ir}$ cm$^{-2}$, commercial IrO2 at 0.25 mg$_{Ir}$ cm$^{-2}$, and Pt/C at 1 mg$_{Pt}$ cm$^{-2}$. The MEA assembly included end plates, bipolar plate [Pt-coated Ti (anode) and graphite (cathode)], a porous transport layer (PTL; Ti felt (2GDL9N-025, Beakart, Belgium) for the anode and carbon paper (39BB, SGL Carbon, Germany) for the cathode), and gaskets. The Ti felt was pretreated with 5% oxalic acid at 60 °C for 30 min prior to use. The MEA was pressed with the PTLs using a force of 1 metric ton for 1 min at 120 °C. Single-cell tests were performed under ambient pressure with a continuous supply of clean distilled water. The active area and cell temperature were 4 cm$^2$ and 80 °C, respectively. Electrochemical analysis was performed using an HCP-803 instrument (Bio-Logic, France). Tests were conducted using activation, LSV, and EIS. Activation was performed using chronoamperometry for 10 min at 1.5 V. LSV measurements were performed at a scan rate of 10 mV s$^{-1}$ in the range of 1.2–2.0 V. EIS measurements were conducted in the frequency range of 10 kHz to 10 mHz at 2.0 V and an amplitude of 10 mV. Stability tests were conducted at a constant current density of 1.0 A cm$^{-2}$.

## Material characterization

TEM and HRTEM analyses were carried out using a TECNAI G2 20 S-twin instrument operating at 200 kV and a TECNAI G2 F30ST instrument at 300 kV, respectively. Aberration-corrected imaging and high-resolution EDS measurements were performed using a Titan Probe Cs TEM (300 kV) with chemi-STEM capabilities at the FEI Nanoport in Eindhoven. EDS elemental mapping utilized a high-efficiency Super-X detection system with an XFEG electron source, incorporating four silicon drift detectors for enhanced detection near the analyzed region. Compared to standard EDS detectors with Schottky FEG sources, the chemi-STEM setup with X-FEG and the Super-X detector improved X-ray generation by up to fivefold and X-ray collection by up to tenfold. All STEM images and compositional maps were obtained using HAADF-STEM. PXRD patterns were measured on a Rigaku Ultima III diffractometer with Cu $K_\alpha$ radiation (graphite monochromatized) at 48 kV and 40 mA. XPS measurements were performed using an ULVAC-PHI X-tool instrument with a monochromatic AI $K_\alpha$ radiation source (1486.6 eV) operating at 24.1 W, referencing the C 1s peak at 284.5 eV.

## In situ XAFS measurements

In situ XANES and EXAFS measurements for the Ru K-edge and Ir L3-edge were performed at the 7D beamline of PLS-II, located at the Pohang Accelerator Laboratory in the Republic of Korea. Monochromatic X-rays were generated using a double-crystal monochromator equipped with Si (111) crystals for the energy scans. The XAS experiments were conducted in a fluorescence-transmission geometry, where fluorescence mode was used to record sample spectra, and the reference material's spectrum was simultaneously measured in transmission mode at room temperature. Calibration of the Ru K-edge and Ir L3-edge XANES spectra was achieved using Ru foil and Ir black powder at 22,117 eV and 11,215 eV, respectively. The raw XAS data were analyzed using the ATHENA software, with the Ru K-edge position determined through the first-derivative method and the Ir L3-edge white-line position identified using the second-derivative method.

## EXAFS measurements

EXAFS data were analyzed following established protocols using the ATHENA module within the IFEFFIT software suite[62]. The $k^3$-weighted EXAFS spectra were obtained by removing the post-edge background from the total adsorption and normalizing it relative to the edge-jump step. The $k^3$-weighted $\chi(k)$ data for the Ru K-edge and Ir L$_3$-edge were then Fourier-transformed into real space using a Hanning window with a $dk$ value of 1.0 Å$^{-1}$, allowing separation of the contributions from

distinct coordination shells. To evaluate the structural environment around the central atoms, least-squares fitting was carried out using the ARTEMIS module in conjunction with the FEFF6 ab initio code. Theoretical models for EXAFS fitting were developed using reference materials, with $RuO_2$ and $IrO_2$ serving as the basis for the Ru–O and Ir–O scattering paths, respectively.

### Differential electrochemical mass spectroscopy (DEMS)

In-situ DEMS using $H_2^{18}O$ was conducted with an HPR-40 quadrupole mass spectrometer system (HIDEN Analytical Limited, England) and Type A cell to investigate the extent of the lattice oxygen mechanism during the OER[58,63]. Catalysts were deposited on polished glassy carbon (GC) electrodes (5 mm in diameter) at a loading of $40\ \mu g\ cm^{-2}$. The setup included a GC electrode as the working electrode, an Ag/AgCl electrode as the reference, and a Pt wire were as the counter electrode. The $^{18}O$ isotope labeling of the catalysts was achieved through 5 cyclic voltammetry (CV) cycles at a scan rate of $5\ mV\ s^{-1}$ in $0.1\ M\ HClO_4$ containing $H_2^{18}O$. During the labeling process, the electrolyte was circulated through the cell at a flow rate of $0.9\ mL\ s^{-1}$. For $RuIrO_2$, the CV potential range was $1.25–1.65\ V_{RHE}$, and for $RuO_2$, it was $1.25–1.95\ V_{RHE}$, ensuring comparable current densities. The electrodes were subsequently rinsed several times with $H_2^{16}O$ to remove any remaining $H_2^{18}O$. Finally, the labeled electrodes were placed in $0.1\ M\ HClO_4$ with $H_2^{16}O$, and CV was performed within the same potential ranges. During the OER, $^{32}O_2$ ($^{16}O^{16}O$) and $^{34}O_2$ ($^{16}O^{18}O$) generated were monitored using mass spectrometry, with baseline correction applied to the signals.

## Data availability

The data that support the findings and conclusions generated in this study are provided in the main article and the Supplementary Information. Source data are provided with this paper.

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

## Acknowledgements

This study was supported by the National Research Foundation of Korea (NRF) (Grant Nos. RS-2023-00256106, 2019R1A6A1A11044070, RS-2024-00466554) and the collaborative research project (R-217194.0001) of Hyundai Motor Company awarded to K.L. Additionally, S.B. was supported by the Carbon Neutral Industrial Strategic Technology Development Program (RS-2023-00261088) funded by the Ministry of Trade, Industry & Energy (MOTIE, Korea) and generous supercomputing time from KISTI. S.J.Y. supported by the NRF (Grant No. 2021M3H4A1A02042948) and the KIST Institutional Program of Korea Institute of Science and Techynology (KIST). T.K. (Taehyun Kwon) was supported by the Core Research Institute (CRI) Program, the Basic Science Research Program through the NRF, Ministry of Education (RS-2017-NR023047). T.K. (Taekyung Kim) and H.B. were supported by the Korea Basic Science Institute (KBSI) (project no. A424100). Experiments conducted at PLS-II were supported in part by MSICT and POSTECH. The authors also thank the KBSI Seoul and Busan Centers for providing access to their HRTEM and XPS instruments.

## Author contributions

Y.P., H.Y.J., T.K.L., and T.K. (Taekyung Kim) contributed equally to this study. Y.P. and K.L. designed the study. Y.P. conducted the synthesis, electrochemical half-cell measurements, and material characterization studies using TEM, PXRD, and XPS. H.Y.J. and S.B. performed DFT calculations. D.K. (Doyeop Kim) and D.K. (Dongjin Kim) synthesized the catalysts. T.K.L. and S.J.Y. constructed the MEA settings for the PEMWE tests. T.K. (Taekyung Kim) and H.B. contributed to HRTEM usage and corresponding data analysis. T.K. (Taehyun Kwon) conducted in situ XAS measurements and analyzed XANES and EXAFS data. T.K. (Taehyun Kwon) and J.C. conducted in situ DEMS measurements. The manuscript was discussed and written by all authors.

## Competing interests

The authors declare no competing interests.
