## [Transparent Peer Review file · Nature Communications]

Atomic-Level Ru-Ir Mixing in Rutile-Type (RuIr)O₂ for Efficient and Durable Oxygen Evolution Reaction

Corresponding Author: Professor Kwangyeol Lee

Version 0:

Reviewer comments:

Reviewer #1

(Remarks to the Author)

The authors integrate OER-36 active Ir atoms into the RuO₂ matrix by leveraging the changeable growth behavior of Ru/Ir 37 atoms on lattice parameter-modulated templates with mixed RuIr alloy to obtain RuIr_xO₂. They achieved (RuIr)O₂/C electrocatalyst at exceptional performance, maintaining its activity for 360 h at 100 mA cm⁻². In PEMWE, the catalysts achieved a superior current density of 4.96 A cm⁻² and mass activity of 19.84 A mgRu+Ir⁻¹ at 2.0 V.

There are a couple of points in the structure analysis needed to be addressed.

In Fig. 5 In-situ XANES analysis, Ru⁴⁺ in RuO₂ and Ir⁴⁺ in IrO₂ are the highest observed oxidation states, no evidence presented to support the statement of "The cyclic voltammetry (CV) curves (Supplementary Fig. 17) exhibited discernible reversible peaks corresponding to changes in the oxidation states of Ru and Ir within (RuIr)O₂/C and RuO₂@IrO₂/C. Remarkably, only the Ir³⁺/Ir⁴⁺ redox peak was observed for RuO₂@IrO₂/C, whereas Ir³⁺/Ir⁴⁺ and Ru³⁺/Ru⁴⁺ peaks were observed for (RuIr)O₂/C,"

In Fig. 7 In-situ EXAFS analysis, didn't see any peaks corresponding to the assignment of Ir-Ir or Ru-Ru or Ir-Ru to provide evidence for the proposed Ir-O-Ru charge transfer pathway.

Reviewer #2

(Remarks to the Author)

This manuscript by Park et al. reports a study on the integration of Ir active atoms into the RuO₂ matrix by leveraging the changeable growth behavior of Ru/Ir atoms on lattice parameter-modulated templates. The resulting (RuIr)O₂/C electrocatalyst exhibited good OER activity and stability. The authors claimed that the catalytic performance arose from the maximized synergy between the Ru and Ir atoms mixed at the atomic level. The Ir atoms stabilized the local coordination environment around Ru, preventing overoxidation of Ru and optimizing the oxygen-intermediate adsorption energy. However, many mechanisms and modification strategies for Ru-Ir oxides to improve the electrocatalytic OER performance have been reported in previous literatures (Nat. Commun. 2023, 14, 5365; ACS Catal. 2021, 11, 15, 9300-9316; Adv. Funct. Mater. 2024, 2402226). The novelty and significance of this work should be highlighted. And some conclusions are not well supported by the present experimental and theoretical results. Especially, more experimental characterizations are needed to verify the structure of (RuIr)O₂/C and the catalytic performance. Therefore, in its present form, I consider that the manuscript cannot meet the high standards of Nature Communications. There are a number of critical issues that need to be resolved.

1. Up to now, the preparation of Ru-Ir oxide catalysts and the corresponding catalytic mechanisms have been widely studied (Nat. Commun. 2023, 14, 5365; ACS Catal. 2021, 11, 15, 9300-9316; Adv. Funct. Mater. 2024, 2402226; ChemCatChem 2024, 16, 2023012; J. Mater. Chem. A, 2023, 11, 25268-25274). Therefore, where are the advance and major breakthrough of the views mentioned by the author in relation to the reported results?

2. The analyses of the XAFS spectra are not convincing.

- a) The XAFS results present average information about the samples. In this work, the lattice-expanded template (e-Ni₃S₄) was prepared by introducing larger-sized Ir atoms into the Ni₃S₄ framework. The Ru-Ir phase was formed on the e-Ni₃S₄ surface. Authors should carefully consider the contribution of Ir in e-Ni₃S₄ when analyzing Ir L₃-edge XAFS spectra.
- b) The authors claimed that the Ni₃S₄-based templates were completely transformed into NiO during the thermal treatment. However, according to Ni K-edge XANES spectra in Supplementary Fig. 15, the oscillations of (RuIr)₂O₂/C and RuO₂@IrO₂/C are completely different from NiO in the range of 8360 to 8440 eV.
- c) For Ru K-edge EXAFS spectra in Fig. 7c, this Ru–O bond elongation (from +1.2 V to +1.45 V) could weaken the electron-withdrawing effect of the neighboring coordinated oxygen atoms. However, the reason for the contraction of the Ru–O bond (from +1.45 V to +1.6 V) was not provided.
- d) For Ru K-edge EXAFS fitting results at 0.4 V, Fig. 7c is not consistent with Supplementary Table 6.

3. The discussion of electrocatalytic characterizations are inadequate.

- a) Ni₃S₄-based templates in the catalysts may have an effect on OER performance. Specifically, Ni-based compounds are difficult to maintain structural stability in strongly acidic media. More analyses should be provided.
- b) The authors should indicate the attribution of reversible peaks in Supplementary Fig. 17 in detail. Could there be Ni related redox peaks?
- c) The SA of (RuIr)₂O₂/C was superior to those of other samples, indicating the superior intrinsic activity of (RuIr)₂O₂/C rather than “the presence of abundant active sites”.
- d) (RuIr)₂O₂/C catalyst contains carbon. Could carbon corrosion under acidic OER condition affect catalytic stability?
- e) The comparison between Supplementary Fig. 21 and Fig. 22 is not convincing, because the line profile analyses depend strongly on the region selected. Supplementary Fig. 21a and Fig. 22a look very similar.
- f) Detailed synthetic method of RuO₂ NPs/C should be described.

4. The authors mentioned that the inner region (part iii) of (RuIr)₂O₂/C showed the (004) and (011(-)) facets indicative of a residual metallic phase. There is no evidence of the existence of (004) in the manuscript.

5. Why is there a large difference in Ru and Ir intensities between Fig. 1i and Supplementary Fig. 6d?

6. According to the results of in-situ XANES, the authors believed that all Ir atoms were placed at the AS, where they exhibited the highest oxidation states. More explanation of this conclusion should be provide.

7. The authors should carefully check the full text and improve the accuracy of the expression. Examples are as follows:

- a) “radius = 112 pm, cf. 110 pm for Ni” in Page 5, what does “cf.” mean?
- b) What does “HFR” mean in Fig. 4e?
- c) “PXRD patterns of Ru NPs/C and RuO₂ NPs/C” in Supplementary Page 18.
- d) “Ru 3p_{3/2} deconvoluted XPS spectra” and “Area ratio of Ru species (%)” in Supplementary Page 29.
- e) The First derivatives of Ru K-edge XANES regions for Ru(acac)₃ in Supplementary Fig. 33 is not clear.

Reviewer #3

(Remarks to the Author)

In this work, the active atom Ir was integrated into the RuO₂ matrix at the atomic scale, and the controlled growth of RuIr₂O₂ was achieved at the lattice-expanded Ni₃S₄ (e-Ni₃S₄), which maximally suppressed the participation of lattice oxygen in the OER reaction and synergized the good stability of Ru and the high activity of Ir. The catalysts were also driven at 174 mV for 10 mA cm⁻² and operated stably for 360 h. XAS analysis and DFT calculations further verified the electron transfer of Ir and Ru, thus promoting the catalyst AEM pathway. In this study, it was found that the synergistic effect generated by atomic-level multi-elements can effectively enhance the acidic OER activity, which lays the foundation for efficient green hydrogen production. It is recommended that this paper be accepted with following modifications :

1. Please depict the growth process of the RuIr alloy in more detail in Scheme 1.
2. Please give the correspondence between the lattice parameters of the template and the Ru/Ir growth in a more intuitive way.
3. XRD comparisons of Ni₃S₄ and e-Ni₃S₄ should be provided
4. Please explain what does the arrows mean in Figure 2(b).
5. Please label the fitted line for the Tafel slope in Figure 3(b).
6. It should be indicated whether the dissolution of Ru and Ir elements occurred after the stability test.
7. Please elaborate on how Ru, Ir synergistically and effectively inhibits the LOM mechanism and promotes the AEM process.
8. Whether the cell voltage was iR-compensated and whether it was necessary.

Version 1:

Reviewer comments:

Reviewer #1

(Remarks to the Author)

The revisions and responses to the review comments are satisfied.

Reviewer #2

(Remarks to the Author)

In the revised manuscript of NCOMMS-24-20531A, the authors have added some experimental and calculational results (XPS, HRTEM, and Bader charge analysis) to support the structure and performance of (RuIr)O₂/C electrocatalyst according to the reviewers' suggestions. The quality of the manuscript has been improved, and now I recommend the publication of this work in Nature Communication. However, the authors should address the following issues.

- 1) The authors claimed that a large fraction of Ni₃S₄ maintained its presence at OER potentials. The reason for the difference in S 2p XPS spectra before and after the 24 h OER operation should be explained (Supplementary Fig. 15 and Fig. R7b).
- 2) The authors were encouraged to add experimental results of SiO₂ as a sacrificial substrate into the Supplementary Information to better illustrate the role of carbon substrate.

Reviewer #3

(Remarks to the Author)

In this research, the integration of active Ir atoms into the RuO₂ matrix by a controlled method can greatly optimize the synergistic OER between Ru and Ir activity centers. As a result, the optimized (RuIr)O₂/C exhibits an excellent overpotential of only 174 mV at 10 mA/cm² and a good stability of 100 mA/cm². In further application to PEMWE, it also demonstrates its ability to operate stably at 1 A/cm². In situ characterization and DFT calculations further illustrate the significant role of the atomic level mixing of Ru and Ir in optimizing the adsorption energy. It is recommended that the manuscript be considered for acceptance with the following revisions.

1. The keywords in the manuscript does not highlight the key points and highlights of the paper, please rethink and revise it.
2. Please describe the unique advantages and rationale for the selection of Ni₃S₄ and e-Ni₃S₄ as templates for atomic level mixing
3. Please give the performance comparison of (RuIr)O₂/C at different current densities (10, 100 mA cm⁻²) in the LSV plot of Fig. 3.
4. Please provide the concentration of Ru and Ir ions in the electrolyte solution after the stability test of different catalysts to demonstrate the reduction of elemental dissolution.
5. Whether catalyst detachment and dissolution occurs during PEMWE testing, especially high-current testing, please identify.

Version 2:

Reviewer comments:

Reviewer #2

(Remarks to the Author)

In the revised manuscript of NCOMMS-24-20531B (Title: Rutile-type (RuIr)O₂ with atomic-level Ru-Ir mixing: Highly efficient and durable electrocatalyst for oxygen evolution reaction), the authors have clarified the question raised in the second round of review, and added some experimental analyses (TEM, elemental mapping, and ICP-MS) according to the reviewers' suggestions. Now I recommend the publication of this work in Nature Communication.

Reviewer #3

(Remarks to the Author)

There is no problem with the revised manuscript, and I recommend its acceptance for publication.

Kwangyeol Lee, Professor

Department of Chemistry

Korea University, Seoul 02841, Republic of Korea

Phone: 82-2-3290-3139

Fax: 82-2-3290-3121

E-mail: kylee1@korea.ac.kr

August 9, 2024

Dear referees

Thank you for your time, effort, and dedication in providing insightful comments on our manuscript entitled, “**Rutile-type (RuIr)O₂ with atomic-level Ru-Ir mixing: Highly efficient and durable electrocatalyst for oxygen evolution reaction**”, submitted to Nature Communications (Research Article, No. NCOMMS-24-20531) Your valuable advice has significantly improved the quality of our paper during the revision process. We have diligently reflected on your detailed suggestions and addressed all the issues and concerns raised by reviewers.

We conducted a series of additional electrocatalytic experiments, XPS, and XAFS analyses to enhance our material characterization and electrochemical investigation. Specifically, we expanded our experimental analysis to cover the phase changes of the Ni₃S₄-based template and provided a detailed discussion of how the adjustment of the lattice parameters of the template influences the growth rate of subsequent metal deposition. These enhancements broaden the scope of our discussion and provide a more in-depth analysis.

This revised manuscript includes further explanations and references to clarify our arguments. Additionally, we have thoroughly checked and corrected any typographical errors throughout the manuscript. We apologize for any inconvenience this might have caused. Here, we provide our point-by-point response to faithfully meet your expectations.

Point-by-point responses to reviewers' comments.

Reviewer #1

The authors integrate OER-36 active Ir atoms into the RuO₂ matrix by leveraging the changeable growth behavior of Ru/Ir 37 atoms on lattice parameter-modulated templates with mixed RuIr alloy to obtain Ru_xIr_yO₂. They achieved (RuIr)O₂/C electrocatalyst at exceptional performance, maintaining its activity for 360 h at 100 mA cm⁻². In PEMWE, the catalysts achieved a superior current density of 4.96 A cm⁻² and mass activity of 19.84 A mg_{Ru+Ir}⁻¹ at 2.0 V. There are a couple of points in the structure analysis needed to be addressed.

We are grateful for your evaluation of our work. We have made every effort to incorporate your valuable suggestions to improve the overall quality of our manuscript. Please find below our point-by-point responses to your comments.

Comments 1: In Fig. 5 In-situ XANES analysis, Ru^{4+} in RuO_2 and Ir^{4+} in IrO_2 are the highest observed oxidation states, no evidence presented to support the statement of “The cyclic voltammetry (CV) curves (Supplementary Fig. 17) exhibited discernible reversible peaks corresponding to changes in the oxidation states of Ru and Ir within $(RuIr)O_2/C$ and $RuO_2@IrO_2/C$. Remarkably, only the Ir^{3+}/Ir^{4+} redox peak was observed for $RuO_2@IrO_2/C$, whereas Ir^{3+}/Ir^{4+} and Ru^{3+}/Ru^{4+} peaks were observed for $(RuIr)O_2/C$,”

Author’s response: We sincerely appreciate the opportunity to clarify the explanation of cyclic voltammetry (CV) curves. To address the reviewer’s comments and to examine the changes in the oxidation states of Ru and Ir in more detail, we remeasured CV curves of $(RuIr)O_2/C$, $RuO_2@IrO_2/C$, and home-made RuO_2 NPs/C by extending the potential range up to 1.5 V_{RHE} . This was necessary because the CV peak corresponding to highly oxidized Ir species ($> 4+$), as observed in in-situ XANES (**Fig. 5**) and the XPS (**Supplementary Figs. 26, 27**) results, appears at potentials above 1.2 V_{RHE} . We then conducted a detailed analysis of the various peaks corresponding to the oxidation states of Ru and Ir observed in these measurements.

As shown in **Fig. R1** and **Table R1**, for $(RuIr)O_2/C$, a distinct redox peak for over-oxidation of Ir appeared at potentials above 1.2 V_{RHE} (*Adv. Funct. Mater.* **2020**, 30, 2003935; *Chem. Sci.* **2015**, 6, 3321). Additionally, in $RuO_2@IrO_2/C$, a redox peak corresponding to Ru^{4+}/Ru^{6+} was observed around 0.9 V_{RHE} , which is not seen in $(RuIr)O_2/C$, indicating that the Ru in $RuO_2@IrO_2/C$ undergoes over-oxidation during the OER operation (*ChemElectroChem* **2015**, 2, 1128; *J. Phys. Chem. C* **2016**, 120, 2562; *J. Electroanal. Chem. Interf. Electrochem.* **1990**, 296, 37; *J. Electroanal. Chem.* **1996**, 415, 89).

To provide a detailed and clear explanation of the changes in oxidation states for $(RuIr)O_2/C$ and $RuO_2@IrO_2/C$ during the OER operation, we have replaced **Supplementary Fig. 17** with **Fig. R1** in the revised Supplementary Information and corrected the discussion in the revised Manuscript to reflect these results.

Fig. R1 Cyclic voltammetry (CV) curves of **a** (RuIr) O_2/C , **b** $RuO_2@IrO_2/C$, and **c** RuO_2 NPs/C measured at a scan rate of 20 mV s^{-1} within a potential range of $0.0 - 1.5 \text{ V}_{RHE}$.

Potential range (V vs. RHE)	
Ru^{3+}/Ru^{4+}	0.6 – 0.8
Ru^{4+}/Ru^{6+}	0.8 – 1.0
Ir^{3+}/Ir^{4+}	0.8 – 1.2
Ir^{4+}/Ir^{5+}	1.2 – 1.5

Table R1. Reference potential ranges for the oxidation peaks of Ru and Ir species in cyclic voltammetry (CV) curves, as investigated in the reported research papers (*Adv. Funct. Mater.* **2020**, 30, 2003935; *Chem. Sci.* **2015**, 6, 3321; *ChemElectroChem* **2015**, 2, 1128; *J. Phys. Chem. C* **2016**, 120, 2562; *J. Electroanal. Chem. Interf. Electrochem.* **1990**, 296, 37; *J. Electroanal. Chem.* **1996**, 415, 89).

Changes Made:

- We have replaced **Supplementary Fig. 17** with **Fig. R1** on page 19 of the revised Supplementary Information.
- We have replaced the discussions for the CV curves in more detail on page 8, line 8 of the revised Manuscript.

“The cyclic voltammetry (CV) curves exhibited discernible reversible peaks corresponding to changes in the oxidation states of Ru and Ir within (RuIr) O_2/C and $RuO_2@IrO_2/C$. As shown in **Supplementary Fig. 17a**, two distinct redox peaks for Ru^{3+}/Ru^{4+} (0.7 V_{RHE}) and Ir^{4+}/Ir^{5+}

(1.2 – 1.5 V_{RHE}) were observed in (RuIr)O₂/C. However, in the case of RuO₂@IrO₂/C (**Supplementary Fig. 17b**), a redox peak corresponding to Ru⁴⁺/Ru⁶⁺ was observed around 0.9 V_{RHE} , which is not seen in (RuIr)O₂/C.^{21,22} These results imply that the Ir in (RuIr)O₂/C undergoes over-oxidation instead of Ru, with rutile Ru⁴⁺ remaining stable, whereas, in RuO₂@IrO₂/C, overoxidized Ru is present but unstable.”

- We have replaced the “*ChemElectroChem* **2**, 1128-1137 (2015)” with “*Chem. Sci.* **6**, 3321-3328 (2015) in **reference 21** of the revised Manuscript.

Comments 2: *In Fig. 7 In-situ EXAFS analysis, didn't see any peaks corresponding to the assignment of Ir-Ir or Ru-Ru or Ir-Ru to provide evidence for the proposed Ir-O-Ru charge transfer pathway.*

Author's response: We sincerely appreciate your valuable comment and insightful suggestion regarding the in-situ EXAFS analysis. To address this, we re-fitted the Ru K-edge FT-EXAFS spectra measured at the applied potential of 0.4 V_{RHE} , considering the Ru metal species, and marked the positions of Ru-M (Ru-Ru or Ru-Ir) bonds on the graph. Additionally, we indicated the positions of Ir-M (Ir-Ir or Ir-Ru) bonds on the Ir L₃-edge EXAFS spectra.

As shown in **Fig. R2**, the peaks observed at the 2-3 Å position correspond to metal-metal bonds (**Supplementary Tables 8, 9**). For Ir, even at an increased potential of 1.6 V_{RHE} , the metal peak at the 2-3 Å position remains. However, the metal-metal bond peak disappeared for Ru when the potential was increased to 1.6 V_{RHE} . This observation is consistent with the in-situ XANES (**Fig. 5**) and XPS (**Supplementary Figs. 26, 27**) results presented in this study. To demonstrate the presence of metal-metal bonds more clearly, we included the metal species in the fitting only for the case at the applied potential of 0.4 V_{RHE} .

Fig. R2 In situ Ir L₃-edge and Ru-K edge FT-EXAFS spectra of (RuIr)O₂/C obtained at different applied potentials in O₂-saturated 0.1 M HClO₄.

Changes Made:

- We have replaced **Fig. 6b, c** with **Fig. R2** on page 39 of the revised Manuscript.

Reviewer #2

This manuscript by Park et al. reports a study on the integration of Ir active atoms into the RuO₂ matrix by leveraging the changeable growth behavior of Ru/Ir atoms on lattice parameter-modulated templates. The resulting (RuIr)O₂/C electrocatalyst exhibited good OER activity and stability. The authors claimed that the catalytic performance arose from the maximized synergy between the Ru and Ir atoms mixed at the atomic level. The Ir atoms stabilized the local coordination environment around Ru, preventing overoxidation of Ru and optimizing the oxygen-intermediate adsorption energy. However, many mechanisms and modification strategies for Ru-Ir oxides to improve the electrocatalytic OER performance have been reported in previous literatures (Nat. Commun. 2023, 14, 5365; ACS Catal. 2021, 11, 15, 9300-9316; Adv. Funct. Mater. 2024, 2402226). The novelty and significance of this work should be highlighted. And some conclusions are not well supported by the present experimental and theoretical results. Especially, more experimental characterizations are needed to verify the structure of (RuIr)O₂/C and the catalytic performance. Therefore, in its present form, I consider that the manuscript cannot meet the high standards of Nature Communications. There are a number of critical issues that need to be resolved.

We sincerely thank you for your great effort in reviewing our manuscript. We also appreciate your careful reading, enlightening comments, and valuable suggestions, which have significantly helped improve our manuscript. In response to the reviewer's questions, particularly regarding the in-situ XAFS analysis results and the electrocatalytic characterization, we have added control experiments and expanded the discussion. We have made every effort to address the comments thoroughly, and our specific replies and modifications are listed as follows:

Comments 1: *Up to now, the preparation of Ru-Ir oxide catalysts and the corresponding catalytic mechanisms have been widely studied (Nat. Commun. 2023, 14, 5365; ACS Catal. 2021, 11, 15, 9300-9316; Adv. Funct. Mater. 2024, 2402226; ChemCatChem 2024, 16, 2023012; J. Mater. Chem. A, 2023, 11, 25268-25274). Therefore, where are the advance and major breakthrough of the views mentioned by the author in relation to the reported results?*

Author's response: Thank you for your comment. We appreciate the opportunity to clarify the novelty and major breakthroughs of our study. While previous studies have primarily focused on the synthesis of simple RuIr alloys and their performance as OER electrocatalysts, our study goes beyond merely demonstrating good performance. Our work delves into the underlying reasons for enhanced OER performance and elucidates the individual contributions of Ru and Ir sites in the alloy. Specifically, we provide a detailed experimental and theoretical analysis of the synergistic effects between Ru and Ir and how these effects can be optimized. Key novelty and breakthroughs of this study are summarized below:

Firstly, we provide detailed mechanistic insights. We have identified the dual role of Ir in the RuIr alloy. The Ir acts as an active site, suppressing the LOM (lattice oxygen oxidation mechanism) and promoting the AEM (adsorbate evolution mechanism), thereby significantly enhancing the OER performance. Additionally, Ir shares electrons with Ru, continuously activating Ru and thereby greatly improving the catalytic activity.

Secondly, our study demonstrates the importance of optimal atomic mixing. We achieved this through a novel synthesis strategy that involves adjusting the lattice parameters of the template. This template not only serves as a synthetic support but also plays a crucial role in the crystal growth of Ru and Ir atoms.

More specifically, by manipulating the lattice parameters of the template, we have developed a method to control the growth rate, degree of mixing, and crystal growth direction of the Ru and Ir atoms. This approach, while commonly used in semiconductor materials such as battery cathodes and anodes, is relatively unexplored in the field of nanocatalysts for water splitting.

Lastly, we have also confirmed that these catalysts exhibit superior performance in practical applications, such as PEMWE, validating the practical significance of our findings.

In summary, our study provides a comprehensive understanding of the Ru-Ir alloy system, detailing the mechanisms that contribute to its high OER activity and stability. We introduce a novel synthesis strategy that leverages template lattice parameter control to optimize the atomic

configuration and growth of the Ru/Ir atoms. This approach represents a significant advancement over previous studies and opens new pathways for the development of high-performance OER electrocatalysts.

Comments 2: *The analyses of the XAFS spectra are not convincing.*

a) The XAFS results present average information about the samples. In this work, the lattice-expanded template (e-Ni₃S₄) was prepared by introducing larger-sized Ir atoms into the Ni₃S₄ framework. The Ru-Ir phase was formed on the e-Ni₃S₄ surface. Authors should carefully consider the contribution of Ir in e-Ni₃S₄ when analyzing Ir L₃-edge XAFS spectra.

Author's response: Thank you for the reviewer's valuable comment. Based on the ICP-AES analysis (**Supplementary Table 5**), we confirmed that the doping level of Ir atoms in the Ni₃S₄ matrix of the e-Ni₃S₄ template is approximately 1.06 wt%, indicating a very low concentration with minimal impact. In response to the reviewer's comments, we intended to conduct an Ir L₃-edge XAFS analysis of e-Ni₃S₄. However, the Pohang Accelerator in South Korea is undergoing long-term repairs, making this measurement challenging. As an alternative, we performed XPS analysis to determine the chemical states of Ni and S in Ni₃S₄ and e-Ni₃S₄ (*Adv. Funct. Mater.* **2019**, 1900315; *J. Mater. Chem. C* **2018**, 6, 1822; *J. Solid State Chem.* **2022**, 315, 123542).

As shown in the XPS results (**Table R2, 3 and Fig. R3**), the Ir dopant did not affect the chemical states of Ni and S in the two templates. Therefore, the small amount of Ir dopant in Ni₃S₄ does not cause any changes in the electronic structure of the template but only alters the lattice parameter, subsequently influencing the growth pattern of the Ru and Ir atoms.

We utilized the Ir dopant solely to induce partial changes in the lattice structure of Ni₃S₄, aiming to control the growth rate of the RuIr shell and the Ru/Ir atomic configuration within the shell. When the pristine Ni₃S₄ was reacted simultaneously with Ru³⁺ and Ir³⁺ via solvothermal synthesis, rapid Ru deposition followed by Ir deposition on the Ni₃S₄ was observed during the early stages of the reaction, resulting in the formation of a Ru@Ir inner shell@outer shell configuration (**Fig. R4a, c, d-f**). However, for the e-Ni₃S₄, the Ir dopant introduced into the Ni₃S₄ matrix caused lattice expansion, which increased the lattice mismatch between Ru and the template, thus reducing the initial rate of Ru deposition. This promotion of atomic-level mixing of Ru and Ir led to the

formation of a RuIr alloy structure (Fig. R4b, c, g-i).

To provide detailed information on the chemical state of Ni₃S₄-based templates, we have newly added a discussion in the revised Manuscript. Additionally, we have included **Table R2**, **Table R3**, and **Fig. R3** for the Ni 2p and S 2p XPS results as **Supplementary Table 1**, **Table 2**, and **Supplementary Fig. 3**, respectively, in the revised Supplementary Information.

Catalyst	Spin state	Oxidation state	Binding energy (eV)
Ni ₃ S ₄	Ni 2p _{3/2}	Ni ²⁺	853.59
		Ni ³⁺	856.34
		Satellite	861.61
e-Ni ₃ S ₄	Ni 2p _{3/2}	Ni ²⁺	853.59
		Ni ³⁺	856.34
		Satellite	861.61

Table R2. Binding energy (eV) used to fit the Ni 2p XPS spectra of Ni₃S₄ and e-Ni₃S₄.

Catalyst	Spin state	Oxidation state	Binding energy (eV)
Ni ₃ S ₄	S 2p	S 2p _{3/2} of S ²⁻	161.98
		S 2p _{1/2} of S ²⁻	163.35
		S ₂ ²⁻	164.60
		SO ₄ ²⁻	169.00
e-Ni ₃ S ₄	S 2p	S 2p _{3/2} of S ²⁻	161.98
		S 2p _{1/2} of S ²⁻	163.35
		S ₂ ²⁻	164.60
		SO ₄ ²⁻	169.00

Table R3. Binding energy (eV) used to fit the S 2p XPS spectra of Ni₃S₄ and e-Ni₃S₄.

[Figure redacted]

Fig. R3 Chemical state of the Ni_3S_4 and $e\text{-Ni}_3\text{S}_4$ templates. **a** Ir 4f, **b** Ni 2p and **c** S 2p XPS spectra of Ni_3S_4 and $e\text{-Ni}_3\text{S}_4$. (*ACS Appl. Mater. Interfaces* **2015**, 7, 4861; *J. Mater. Chem. C* **2018**, 6, 1822; *J. Mater. Chem. A* **2017**, 5, 20985).

Fig. R4 Schematic illustration of Ru and Ir growth on **a** Ni_3S_4 and **b** $e\text{-Ni}_3\text{S}_4$, which afforded Ru@Ir and RuIr shell configurations, respectively. Green, red, yellow, and purple spheres denote Ni, S, Ru, and Ir atoms, respectively. **c** Time-dependent atomic composition analysis obtained by EDS during the deposition of Ru/Ir on Ni_3S_4 and $e\text{-Ni}_3\text{S}_4$. PXRD patterns of **d** $\text{Ni}_3\text{S}_4@RuIr$ and **e** $e\text{-Ni}_3\text{S}_4@RuIr$ obtained at reaction times of 5, 10, and 30 min. HAADF-STEM images of **e** $\text{Ni}_3\text{S}_4@RuIr$ and **h** $e\text{-Ni}_3\text{S}_4@RuIr$ obtained at reaction times of 5, 10, 30, and 60 min (scale bar = 10 nm). Line profile analysis of **f** $\text{Ni}_3\text{S}_4@RuIr$ and **i** $e\text{-Ni}_3\text{S}_4@RuIr$ determined along the lines marked by arrows in **e** and **h**, respectively.

Changes Made:

- We have newly added **Fig. R3** as **Supplementary Fig. 3** on page 4 of the revised Supplementary Information.

- We have newly added **Table R2 and Table R3** as **Supplementary Table 1 and Table 2**, respectively, on page 54-55 of the revised Supplementary Information.
- We have newly added discussion for the chemical state of Ni₃S₄-based templates on page 5, line 18 of the revised Manuscript.

“Moreover, the X-ray photoelectron spectroscopy (XPS) analysis for Ni 2p and S 2p showed that the Ir dopant does not cause any changes in the electronic structure of the template but only alters the lattice structure (Supplementary Fig. 3 and Supplementary Tables 1, 2).”

b) The authors claimed that the Ni₃S₄-based templates were completely transformed into NiO during the thermal treatment. However, according to Ni K-edge XANES spectra in Supplementary Fig. 15, the oscillations of (RuIr)O₂/C and RuO₂@IrO₂/C are completely different from NiO in the range of 8360 to 8440 eV.

Author’s response: We apologize for the confusion caused by the incorrect discussion regarding the Ni K-edge XANES spectra. To address the reviewer’s concerns, we conducted additional Ni 2p and S 2p XPS analyses to examine the phase transform behavior of the Ni₃S₄ phase within (RuIr)O₂/C and RuO₂@IrO₂/C during thermal oxidation (*Adv. Funct. Mater.* **2019**, 1900315; *J. Mater. Chem. C* **2018**, 6, 1822; *J. Solid State Chem.* **2022**, 315, 123542).

Compared to the Ni 2p XPS results of the Ni₃S₄-based template (**Fig. R3**), (RuIr)O₂/C and RuO₂@IrO₂/C exhibited a significant increase in the proportion of Ni²⁺ and satellite peaks, indicating the presence of nickel-oxygen species (*Electrochim. Acta* **1992**, 37, 2029; *ACS Catal.* **2017**, 7, 229), while some Ni³⁺ species still remained. These results imply that the Ni₃S₄ phase in the core of (RuIr)O₂/C and RuO₂@IrO₂/C is partially oxidized to NiO and partially retained as Ni₃S₄ during the thermal oxidation process (**Fig. R5a**). The S 2p XPS spectra also showed the presence of a small amount of S atoms even after thermal oxidation (**Fig. R5b**). Additionally, the ICP-AES analysis (**Supplementary Table 5**) demonstrated the coexistence of the Ni₃S₄ and NiO phases within (RuIr)O₂/C and RuO₂@IrO₂/C.

Therefore, the discussion that the Ni₃S₄-based templates were completely oxidized to NiO during the thermal treatment process is incorrect. To clarify the discussion on the chemical states

of Ni species in (RuIr)O₂/C and RuO₂@IrO₂/C, we have revised the incorrect discussion in the revised Manuscript. Moreover, we have included additional data (**Fig. R5**) in **Supplementary Fig. 15** and provided a detailed discussion on the phase transformation behavior of the Ni₃S₄-based core during thermal oxidation in the **Supplementary Note 3** of the revised Supplementary Information.

Fig. R5 Chemical state of the Ni₃S₄-based templates within (RuIr)O₂/C and RuO₂@IrO₂/C after thermal oxidation. **a** Ni 2p XPS spectra of (RuIr)O₂/C and RuO₂@IrO₂/C. **b** Area percentage of Ni species in Ni₃S₄-based templates (obtained by **Fig. R3a**) and (RuIr)O₂/C, RuO₂@IrO₂/C (obtained by **Fig. R5a**). The satellite peaks indicate the presence of nickel-oxygen species. **c** S 2p XPS spectra of (RuIr)O₂/C and RuO₂@IrO₂/C.

Changes Made:

- We have added **Fig. R5** to **Supplementary Fig. 15** to exhibit the chemical state changes of Ni₃S₄-based templates during thermal oxidation in more detail on page 17 of the revised Supplementary Information.
- We have newly added **Supplementary Note 3** on page 17 of the revised Supplementary Information.
- We have revised the discussion of the phase transformation of Ni₃S₄ during thermal oxidation on page 7, line 24 of the revised Manuscript.

“In all cases, the Ni₃S₄-based templates were partially oxidized to NiO and partially retained as Ni₃S₄ during the thermal treatment (**Supplementary Fig. 15** and **Supplementary Note 3**).”

c) For Ru K-edge EXAFS spectra in Fig. 7c, this Ru–O bond elongation (from +1.2 V to +1.45 V) could weaken the electron-withdrawing effect of the neighboring coordinated oxygen atoms. However, the reason for the contraction of the Ru–O bond (from +1.45 V to +1.6 V) was not provided.

Author’s response: We thank the reviewer for the valuable question, and we are pleased to clarify this issue. In addition to the observed Ru–O bond elongation, we conducted further analysis on the contraction of Ru–O bond (**Fig. R6**). The contraction of the Ru–O bond from +1.45 V_{RHE} to +1.6 V_{RHE} can be attributed to the deprotonation of the HO_{2c} species bonded to Ru. As the potential increases, the HO_{2c} species undergoes deprotonation, leading to a shortening of the Ru–O bond.

To provide a more comprehensive understanding of the changes in Ru–O bond length within (RuIr)O₂/C at applied potentials, we have added a detailed discussion in the revised Manuscript and replaced **Supplementary Fig. 35** with **Fig. R6** in the revised Supplementary Information.

Fig. R6 Structural model for DFT calculations. The local structures of Ru in the M_{6c} surface structure with the lowest energy shown as 32 in **Supplementary Fig. 42**. M–O bond length values are presented with the change in parenthesis, where blue and red bold fonts indicate an increase and decrease of 0.02 Å or more in M–O bond lengths, respectively.

Changes Made:

- We have replaced **Supplementary Fig. 35** with **Fig. R6** on page 37 of the revised

Supplementary Information.

- We have newly added discussion on page 14, line 14 of the revised Manuscript.

“Additionally, as deprotonation progressed up to +1.6 V_{RHE}, contraction of the Ru-O bond was observed. Theoretical analysis suggested that this is due to the deprotonation of HO_{2c} bonded to Ru at oxidative potentials, resulting in the formation of O_{2c}.”

d) For Ru K-edge EXAFS fitting results at 0.4 V, Fig. 7c is not consistent with Supplementary Table 6.

Author’s response: We apologize for the error. To address the reviewer’s comment, we re-fitted the Ru K-edge FT-EXAFS spectra measured at the applied potential of 0.4 V_{RHE}, considering the Ru metal species.

Fig. R2 In situ Ir L₃-edge and Ru-K edge FT-EXAFS spectra of (RuIr)O₂/C obtained at different applied potentials in O₂-saturated 0.1 M HClO₄.

Changes Made:

- We have replaced **Fig. 6b, c** with **Fig. R2** on page 39 of the revised Manuscript.

Comments 3: *The discussion of electrocatalytic characterizations are inadequate.*

a) Ni₃S₄-based templates in the catalysts may have an effect on OER performance. Specifically, Ni-based compounds are difficult to maintain structural stability in strongly acidic media. More analyses should be provided.

Author's response: We sincerely appreciate your valuable comment and insightful suggestion. We understand the concern regarding the potential dissolution of Ni and S during OER operation in strongly acidic media. To address this, we conducted further analyses to investigate the structural stability of Ni₃S₄ templates under OER conditions.

Our results showed that the Ni₃S₄ phase partially transforms into NiO during thermal treatment, but a significant portion of Ni₃S₄ remains intact (**Fig. R5**). This NiO component tends to dissolve, while a substantial amount of Ni₃S₄ persists during the OER operation (**Fig. R7**). This observation was confirmed through XPS and elemental mapping analyses, which showed the presence of considerable Ni and S even after extended OER operation. As shown in **Fig. R7a, b**, the XPS spectra revealed that the Ni 2p and S 2p signals corresponding to Ni₃S₄ are still detectable, indicating that a large fraction of Ni₃S₄ maintained its presence despite the dissolution of NiO at OER potentials. Additionally, the elemental mapping and line profile analysis further support the existence of residual Ni and S in the catalyst structure (**Fig. R7c-f**).

Despite the partial dissolution of the Ni-based core, the high ratio of Ru and Ir compared to Ni and S in (RuIr)O₂/C and RuO₂@IrO₂/C before OER operation (**Supplementary Table 5**) and the considerable thickness of the Ru/Ir shell grown on the template (**Fig. R8**) ensured that the structure remained intact after the OER.

Furthermore, during thermal oxidation and OER operation, the Ni-based core in the (RuIr)O₂/C and RuO₂@IrO₂/C exhibits similar behavior. This suggests that the Ni-based core does not influence the catalytic performance difference between (RuIr)O₂/C and RuO₂@IrO₂/C. Therefore, the Ni₃S₄-based template is only used as a sacrificial template to modulate the Ru/Ir atomic configuration by altering the growth rate of Ru/Ir atoms.

Fig. R7 Characterization of Ni and S species within $(\text{RuIr})\text{O}_2/\text{C}$ and $\text{RuO}_2@\text{IrO}_2/\text{C}$ after 24 h OER operation. **a** Ni 2p and **b** S 2p XPS spectra of $(\text{RuIr})\text{O}_2/\text{C}$ and $\text{RuO}_2@\text{IrO}_2/\text{C}$ after the OER. HAADF-STEM and elemental mapping images of Ni (green) and S (red) content in **c** $(\text{RuIr})\text{O}_2/\text{C}$ and **e** $\text{RuO}_2@\text{IrO}_2/\text{C}$ after the OER. The line profile analysis of the marked white arrow in panels **c** and **e** for **d** and **f**, respectively.

Fig. R8 Comparison of particle thickness between Ni_3S_4 -based nanoparticles (before RuIr growth) and Ru/Ir oxide-based nanoparticles (after RuIr growth) by counting 100 particles for each sample. *b) The authors should indicate the attribution of reversible peaks in Supplementary Fig. 17 in detail. Could there be Ni related redox peaks?*

Author's response: Thank you for your valuable comment regarding the cyclic voltammetry (CV) curves. To address the reviewer's comment and to examine the changes in the oxidation states of Ru and Ir in more detail, we remeasured CV curves of (RuIr)O₂/C, RuO₂@IrO₂/C, and home-made RuO₂ NPs/C by extending the range up to 1.5 V_{RHE} (**Fig. R1**). This was necessary because the CV peak corresponding to highly oxidized Ir species (>4+) observed in in-situ XANES (**Fig. 5**) and XPS (**Supplementary Figs. 26, 27**) appears at potentials above 1.2 V_{RHE}.

As shown in **Fig. R1** and **Table R1**, for (RuIr)O₂/C, a distinct redox peak for over-oxidation of Ir appeared at potentials above 1.2 V_{RHE} (*Adv. Funct. Mater.* **2020**, 30, 2003935; *Chem. Sci.* **2015**, 6, 3321). Additionally, in RuO₂@IrO₂/C, a redox peak corresponding to Ru⁴⁺/Ru⁶⁺ was observed around 0.9 V_{RHE}, which is not seen in (RuIr)O₂/C, indicating that the Ru in RuO₂@IrO₂/C undergoes over-oxidation during the OER operation (*ChemElectroChem* **2015**, 2, 1128; *J. Phys. Chem. C* **2016**, 120, 2562; *J. Electroanal. Chem. Interf. Electrochem.* **1990**, 296, 37; *J. Electroanal. Chem.* **1996**, 415, 89).

CV is primarily a technique used to investigate the electrochemical properties of the surface of a catalyst. It provides information about the redox processes occurring on the catalyst surface by measuring the current response to an applied voltage. Therefore, due to the significant thickness of the RuIr shell on the Ni₃S₄-based template (**Fig. R8**), the redox peaks corresponding to Ni are difficult to observe in the CV graphs of (RuIr)O₂/C and RuO₂@IrO₂/C. Instead, we analyzed the chemical properties of the Ni₃S₄-based core within (RuIr)O₂/C and RuO₂@IrO₂/C using XPS (**Fig. R5**) and ICP-AES (**Supplementary Table 5**) analysis.

To provide a detailed and clear explanation of the changes in oxidation states for (RuIr)O₂/C and RuO₂@IrO₂/C during OER operation, we have replaced **Supplementary Fig. 17** with **Fig. R1** in the revised Supplementary Information and corrected the discussion in the revised Manuscript to reflect this information.

Fig. R1 Cyclic voltammetry (CV) curves of **a** (RuIr) O_2/C , **b** $RuO_2@IrO_2/C$, and **c** RuO_2 NPs/C measured at a scan rate of 20 mV s^{-1} within a potential range of $0.0 - 1.5 \text{ V}_{RHE}$.

Potential range (V vs. RHE)	
Ru^{3+}/Ru^{4+}	0.6 – 0.8
Ru^{4+}/Ru^{6+}	0.8 – 1.0
Ir^{3+}/Ir^{4+}	0.8 – 1.2
Ir^{4+}/Ir^{5+}	1.2 – 1.5

Table R1. Reference potential ranges for the oxidation peaks of Ru and Ir species in cyclic voltammetry (CV) curves, as investigated in the reported research papers (*Adv. Funct. Mater.* **2020**, 30, 2003935; *Chem. Sci.* **2015**, 6, 3321; *ChemElectroChem* **2015**, 2, 1128; *J. Phys. Chem. C* **2016**, 120, 2562; *J. Electroanal. Chem. Interf. Electrochem.* **1990**, 296, 37; *J. Electroanal. Chem.* **1996**, 415, 89).

Changes Made:

- We have replaced **Supplementary Fig. 17** with **Fig. R1** on page 19 of the revised Supplementary Information.
- We have replaced the discussions for the CV curves in more detail on page 8, line 8 of the revised Manuscript.

“The cyclic voltammetry (CV) curves exhibited discernible reversible peaks corresponding to changes in the oxidation states of Ru and Ir within (RuIr) O_2/C and $RuO_2@IrO_2/C$. As shown in **Supplementary Fig. 17a**, two distinct redox peaks for Ru^{3+}/Ru^{4+} (0.7 V_{RHE}) and Ir^{4+}/Ir^{5+}

(1.2 – 1.5 V_{RHE}) were observed in (RuIr)O₂/C. However, in the case of RuO₂@IrO₂/C (**Supplementary Fig. 17b**), a redox peak corresponding to Ru⁴⁺/Ru⁶⁺ was observed around 0.9 V_{RHE}, which is not seen in (RuIr)O₂/C.^{21,22} These results imply that the Ir in (RuIr)O₂/C undergoes over-oxidation instead of Ru, with rutile Ru⁴⁺ remaining stable, whereas, in RuO₂@IrO₂/C, overoxidized Ru is present but unstable.”

- We have replaced the “*ChemElectroChem* **2**, 1128-1137 (2015)” with “*Chem. Sci.* **6**, 3321-3328 (2015) in **reference 21** of the revised Manuscript.

c) The SA of (RuIr)O₂/C was superior to those of other samples, indicating the superior intrinsic activity of (RuIr)O₂/C rather than “the presence of abundant active sites”.

Author’s response: We apologize for the errors in the discussion regarding this point. The term ‘abundant active sites’ refers to the electrochemical active surface area (ECSA) of catalysts, while specific activity refers to the activity per unit area. Therefore, as the reviewer mentioned, it would be appropriate to state that ‘(RuIr)O₂/C has high intrinsic activity’.

As shown in **Supplementary Fig. 19**, both (RuIr)O₂/C and RuO₂@IrO₂/C have a large surface area due to the dendritic form of RuIr cactus on their surface, resulting in a similar number of active sites and, therefore, comparable ECSA values. However, the specific activity (**Fig. 3c and Supplementary Fig. 20**), which is active per unit area, is much higher for (RuIr)O₂/C than for RuO₂@IrO₂/C, indicating that the intrinsic activity of (RuIr)O₂/C is significantly greater. We have revised the Manuscript to reflect this clarification and corrected the discussion accordingly.

Changes Made:

- We have revised the discussion for specific activities of electrocatalysts on page 9, line 6 of the revised Manuscript.

“The SA of (RuIr)O₂/C (187 A cm_{ECSA}⁻²) exceeded those of other samples, indicative of its superior intrinsic activity (**Fig. 3c, right and Supplementary Fig. 20**).”

d) (RuIr)O₂/C catalyst contains carbon. Could carbon corrosion under acidic OER condition affect catalytic stability?

Author's response: We appreciate the reviewer's insightful question regarding the potential impact of carbon corrosion during the OER operation. The electrocatalysts reported in this study undergo thermal oxidation at 400 °C for conversion to oxide phases, requiring a substrate to prevent nanoparticle aggregation. Therefore, many OER studies use carbon substrates to prevent aggregation during thermal treatment and to enhance performance during electrochemical evaluations (*Nature Materials* **2023**, 22, 100; *Energy Environ. Sci.* **2022**, 15, 1119).

To address this concern, we conducted additional experiments using SiO₂ as a sacrificial substrate instead of carbon (**Fig. R9**). This approach allows us to prepare unsupported RuO₂@IrO₂ and (RuIr)O₂ catalysts, thereby eliminating the potential impact of carbon corrosion during the OER operation and enabling the assessment of OER performance effects from the Ru/Ir atom configuration in both catalysts.

In these experiments, nanoparticles were supported on SiO₂, subjected to thermal oxidation, and subsequently, the SiO₂ was removed using hydrofluoric acid (**Fig. R9a**). RuO₂@IrO₂ and (RuIr)O₂ were synthesized without a supporting substrate, and during the removal process of the SiO₂ substrate, some particle aggregation was observed, which suggests the particle aggregation prevention effect of the carbon support used previously. (**Fig. R9b**). The PXRD results confirmed that both electrocatalysts were successfully converted to their oxide phases during the thermal oxidation process, showing the same behavior as when using a carbon substrate (**Fig. R9c**). Moreover, the Ru and Ir atomic configurations in RuO₂@IrO₂ (**Fig. R10**) (RuIr)O₂ (**Fig. R11**) exhibited the alloy and inner shell@outer shell structures, respectively. This indicates that we successfully synthesized catalysts with the same atomic configurations as those obtained using the SiO₂-based synthesis method.

As shown in **Fig. R12**, the unsupported (RuIr)O₂/C still exhibited enhanced OER activity and stability compared to unsupported RuO₂@IrO₂/C, although the overall performance slightly decreased compared to using a carbon substrate. While the carbon substrate improves the overall electrochemical performance by preventing nanoparticle aggregation and providing better electrical conductivity, the primary factor for the high performance remains the RuIr atomic interaction in (RuIr)O₂/C.

Fig. R9 a Schematic illustration and **b** TEM images for the formation of unsupported $\text{RuO}_2@\text{IrO}_2$ and $(\text{RuIr})\text{O}_2$. **c** PXRD patterns for unsupported $\text{RuO}_2@\text{IrO}_2$ and $(\text{RuIr})\text{O}_2$. Gray boxes denote the remaining metallic species in unsupported $\text{RuO}_2@\text{IrO}_2$ and $(\text{RuIr})\text{O}_2$, which show the same behavior as when using a carbon support.

Fig. R10 a Combined and individual EDS elemental mapping images of O (cyan), Ru (yellow), and Ir (purple) within unsupported RuO₂@IrO₂. The three arrows indicate the range of the line profile in Fig. R11b-d. **b-d** Line profile analysis corresponding to the marked area indicated by the white arrows in panel a.

Fig. R11 a Combined and individual EDS elemental mapping images of O (cyan), Ru (yellow), and Ir (purple) within unsupported RuIrO₂. The three arrows indicate the range of the line profile

in Fig. R11b-d. **b-d** Line profile analysis corresponding to the marked area indicated by the white arrows in panel **a**.

Fig. R12 (a) OER polarization curves of unsupported (RuIr)O₂ and RuO₂@IrO₂. (b) Chronopotentiometry test of unsupported (RuIr)O₂ at 10 mA cm⁻².

e) The comparison between Supplementary Fig. 21 and Fig. 22 is not convincing, because the line profile analyses depend strongly on the region selected. Supplementary Fig. 21a and Fig. 22a look very similar.

Author's response: Thank you for the reviewer's helpful comment on structural characterization after the OER operation. We understand the concern regarding the reliability of the line profile analyses due to the region selected. To address this, we have included additional TEM images taken at lower magnification to provide a more comprehensive view of the structural changes in the nanoparticles after 24 hours of OER operation.

As shown in **Fig. R13**, the structure of the (RuIr)O₂/C remains stable after OER operation, whereas RuO₂@IrO₂/C exhibits significant structural changes. In particular, the dendritic shell composed of Ru/Ir in RuO₂@IrO₂/C shows signs of detachment and thinning due to Ru/Ir dissolution. This observation is not limited to a single particle, as shown in **Supplementary Fig. 22**, but is consistent across most particles examined, as shown in the additional TEM images in **Fig. R13**.

Furthermore, the structural stability of (RuIr)O₂/C and the dendrite detachment in RuO₂@IrO₂/C during the OER are also supported by other analyses. HRTEM images and corresponding FFT patterns show that (RuIr)O₂/C maintained its crystallinity, while

$\text{RuO}_2@\text{IrO}_2/\text{C}$ deteriorated due to Ru and Ir leaching (**Supplementary Fig. 24**). Additionally, ICP-MS analysis confirmed negligible leaching of Ru and Ir from $(\text{RuIr})\text{O}_2/\text{C}$, whereas $\text{RuO}_2@\text{IrO}_2/\text{C}$ exhibited considerable loss of these elements during long-term OER operation (**Supplementary Fig. 25**). These findings indicate that the structural stability of $(\text{RuIr})\text{O}_2/\text{C}$ is maintained post-OER, while $\text{RuO}_2@\text{IrO}_2/\text{C}$ suffers from structural degradation, corroborating the line profile analyses.

To provide a detailed view of the structural destabilization of $\text{RuO}_2@\text{IrO}_2/\text{C}$ during the OER, we have added **Fig. R13** to **Supplementary Fig. 21**. Additionally, we have highlighted the areas where dissolution occurred within the $\text{RuO}_2@\text{IrO}_2/\text{C}$ nanoparticles (**Fig. R14**) and replaced **Supplementary Fig. 23** with **Fig. R14** to clearly show the internal structural changes.

Fig. R13 TEM images of **a** $(\text{RuIr})\text{O}_2/\text{C}$ and **b** $\text{RuO}_2@\text{IrO}_2/\text{C}$ after the OER operation, taken at the various locations on the TEM grids.

Fig. R14 HAADF-STEM images of RuO₂@IrO₂/C after the OER operation. The areas marked with red dashed lines indicate the dissolution site of Ru and Ir atoms.

Changes Made:

- We have added **Fig. R13** to **Supplementary Fig. 21** on page 23 of the revised Supplementary Information.
- We have replaced **Supplementary Fig. 23** with **Fig. R14** on page 25 of the revised Supplementary Information.

f) Detailed synthetic method of RuO₂ NPs/C should be described.

Author's response: We thank the reviewer for pointing out the missing information on the synthetic methods for RuO₂ NPs/C. To reflect the reviewer's comment, we have included the synthetic procedure for home-made RuO₂ NPs/C from Ru NPs/C via a thermal oxidation process in the experimental section of the revised Manuscript.

Changes Made:

- We have newly added a synthetic method for home-made RuO₂ NPs/C on page 19, line 7 of the revised Manuscript.

“Synthesis of home-made RuO₂ NPs/C. The home-made RuO₂ nanoparticles (NPs) were synthesized using a previously reported method with minor modifications^{14,16}. To prepare home-made RuO₂ NPs, Ru NPs were synthesized first. A slurry of Ru(acac)₃ (0.06 mmol), CTAC (0.01 mmol), 1,2-hexadecanediol (0.4 mmol), and OAm (10 mmol) was prepared in a

100 mL Schlenk tube equipped with a magnetic stirrer. The slurry was placed under vacuum in an oil bath at 30 °C for 10 min and charged with Ar (1 atm). Thereafter, the tube was transferred to a preheated oil bath and held at 280 °C for 2 h. The product was sequentially washed with ethanol (15 mL) and toluene (10 mL) and collected by centrifugation. The resulting Ru NPs were supported on carbon black and thermally annealed at 400 °C for 2 h in a flow of N₂-balanced 40% O₂ (0.5 mL min⁻¹) inside a tube furnace, which was identical to the experimental method described above to prepare (RuIr)O₂/C and RuO₂@IrO₂/C.”

Comments 4: *The authors mentioned that the inner region (part iii) of (RuIr)O₂/C showed the (004) and (011(-)) facets indicative of a residual metallic phase. There is no evidence of the existence of (004) in the manuscript.*

Author’s response: We apologize for the confusion caused regarding the lattice structure analysis for (RuIr)O₂/C. We observed the presence of both oxide and metal phases in (RuIr)O₂/C in HAADF-STEM (**Fig. 2d**) and HRTEM (**Supplementary Fig. 10**). In the metallic region of part iii in **Fig. 2d**, the FFT pattern corresponding to the (011̄) facet was observed, while the FFT pattern corresponding to the (004) facet was seen in the HRTEM.

In response to the reviewer’s comment, we have added information on the specific facets corresponding to the observed FFT patterns in HRTEM images (**Fig. R15**). Additionally, the specific facets of the oxide and metal phases observed in the HAADF-STEM and HRTEM images are summarized in **Table R4**.

To provide detailed facet information for HRTEM analysis, we have replaced **Supplementary Fig. 10** with **Fig. R15** in the revised Supplementary Information and corrected the discussion in the revised Manuscript to reflect this information.

Fig. R15 a Enlarged HRTEM image of (RuIr)O₂/C with corresponding FFT patterns for **b** oxide phase and **c** metal phase.

Crystal phase	Facet	d-spacing values (reference)	d-spacing values (measured)	Figure number
Oxide	($\bar{1}10$)	0.317 nm	0.320 nm	α, β
	($2\bar{2}0$)	0.158 nm	0.159 nm	α
	($01\bar{1}$)	0.256 nm	0.259 nm	α, β
	($10\bar{1}$)	0.256 nm	0.258 nm	α, β
	($0\bar{2}0$)	0.224 nm	0.224 nm	β
Metal	(004)	0.107 nm	0.108 nm	α
	($01\bar{1}$)	0.235 nm	0.236 nm	β

α : in Supplementary Fig. 10; β : in Fig. 2

Table R4. Comparison of reference and measured d-spacing values for all facets of both oxide and metal phase present in the (RuIr)O₂/C.

Changes Made:

- We have replaced **Supplementary Fig. 10** with **Fig. R15** on page 12 of the revised

Supplementary Information.

- We have revised the sentence on page 7, line 7 of the revised Manuscript.

“The lattice spacings of 0.259, 0.258, and 0.320 nm, corresponding to rutile-type oxide (01 $\bar{1}$), (10 $\bar{1}$), and ($\bar{1}$ 10) facets, respectively, were observed in the outer regions of (RuIr)O₂/C by HAADF-STEM imaging and FFT analysis (parts i and ii of Fig. 2d and Supplementary Fig. 10b). In contrast, the inner region of (RuIr)O₂/C showed the (01 $\bar{1}$) (part iii of Fig. 2d) and (004) (Supplementary Fig. 10c) facets indicative of a residual metallic phase.”

Comments 5: *Why is there a large difference in Ru and Ir intensities between Fig. 1i and Supplementary Fig. 6d?*

Author’s response: Thank you for pointing out the discrepancy in Ru and Ir intensities between **Fig. 1i** and **Supplementary Fig. 6d**. Addressing your comment, we have discovered that **Supplementary Fig. 6d** was included in error and did not accurately represent the data intended for comparison.

To correct this mistake, we have removed **Supplementary Fig. 6d** from the Manuscript. We apologize for any confusion this might have caused. The correct data for the Ru and Ir intensities are accurately shown in **Fig. 1i**, which has been validated and is representative of our results.

We have revised the Manuscript and the Supplementary Information accordingly, ensuring that all figures and data presented are accurate and consistent. Thank you for your understanding and for bringing this issue to our attention.

Changes Made:

- We have removed **Supplementary Fig. 6d** of the original Supplementary Information.

Comments 6: *According to the results of in-situ XANES, the authors believed that all Ir atoms were placed at the AS, where they exhibited the highest oxidation states. More explanation of this conclusion should be provided.*

Author's response: We appreciate the reviewer's helpful comment. In response, we conducted additional examinations of the electronic properties of various $\text{Ru}_x\text{Ir}_{1-x}\text{O}_2$ structures, specifically focusing on the location of the Ir. The analysis details have been added to the revised Supplementary information.

To model the most realistic surface structures of $(\text{RuIr})\text{O}_2$, we performed the Bader charge analysis based on the locations of Ir atoms. We identified the Ir site with the highest oxidation state for up to three Ir atoms. For a single Ir atom in the system, we considered various sites, including active sites (AS), six-coordinated metal sites (M_{6c}), sub-positions of active sites (sub-AS), and sub-positions of six-coordinated metal sites (sub- M_{6c}). The Ir atom at the AS demonstrated the highest oxidation state. Using this structure as a reference, we compared the average oxidation state of Ir atoms when the second Ir atom was placed at all possible sites. The average oxidation state of Ir atoms was also highest when the second Ir atom was located in the AS. Finally, when two Ir atoms were positioned at the AS, placing the third Ir atom at the AS resulted in the highest average oxidation state for the Ir atoms (**Fig. R16**). This result indicates that the Ir position at the AS is capable of describing the highest oxidation state of Ir observed in **Fig 5**.

Fig. R16 DFT calculations for Bader charges. The Bader charge values of Ir atoms are shown according to their locations in the top layer when the number of Ir atoms is **a** one, **b** two, and **c** three. Ir atoms specified as ‘(AS)’ refer to Ir fixed in the AS position. In the atomic structures, the red balls and yellow and purple polyhedrons represent O, Ru, and Ir atoms, respectively.

The origin of this result can be understood through the Bader charge values of the oxygen atoms in the top layer. There are five types of oxygen in the top layer, labeled O-1 through O-5 (**Fig. R17**). The absolute Bader charge values for O-3 (-0.86), O-4 (-0.89), and O-5 (-0.88) are negative, indicating that these oxygen atoms withdraw a significant amount of electrons from their neighbors. To determine the charge withdrawn by each oxygen atom from a single metal atom, we normalized the absolute Bader charge of the oxygen by the number of its M-O bonds. We found that O-1, which is bonded to the AS, has the most negative normalized Bader charge value (-0.41), indicating a strong tendency to oxidize. This suggests that the metal atom located in the AS is the most oxidized. Therefore, we used the atomic structure with Ir occupying the AS throughout this work.

Fig. R17 The absolute and normalized Bader charge values of each oxygen in the top layer. The normalized values are based on the number of M-O bonds of the metal atom. O-1 and O-5 are bonded in the +z and -z directions relative to the AS, respectively, while O-2 and O-4 are bonded in the +z and -z directions relative to the M_{6c} , respectively. O-3 is located between AS and M_{6c} , bonded to both. O-1 has one M-O bond, O-2 has two M-O bonds, and O-3, O-4, and O-5 each have three M-O bonds. The red balls and yellow and purple polyhedrons in the atomic structures correspond to O, Ru and Ir atoms.

We observed that the oxidation state of Ir increases more significantly than that of Ru as the potential increases (**Fig. 5**). Previous studies have reported that the rutile surface undergoes a transition from OH^* to O^* termination upon the application of the oxidation potential (*Energy Environ. Sci.* **2017**, 10, 2626). Thus, we constructed all possible unique 6 structures satisfying Ru:Ir ratio of 1:1 in the top layer (**Fig. R18a**) and calculated the Bader charges of metal atoms for

both OH* and O* surface terminations. In the OH* termination, Ir and Ru in the top layer had similar Bader charge values (**Fig. R18b**). However, in the O* termination, Ru exhibited a higher Bader charge except for the case of 3Ir_{AS}. Notably, during the transition from OH* to O* termination, Ir showed a greater increase in Bader charge compared to Ru only in the 3 Ir_{AS} structure (**Fig. R19**). This indicates that 3Ir_{AS} is the only case that matches the in situ XANES results, showing a greater increase in the oxidation state of Ir compared to Ru (**Fig. 5**).

Fig. R18 a All possible unique structures satisfying Ru:Ir ratio of 1:1 in the top layer with OH* and O* termination. The names of the structures are designated based on the number of Ir atoms in the AS

b The Bader charge values of Ir and Ru for the OH* and O* terminations of the structures represented in **a**. The red and white balls and yellow and purple polyhedrons in the atomic structures correspond to O, H, Ru and Ir atoms.

Fig. R19 The increase in Bader charge of Ir and Ru during the transition from OH* to O* termination.

Changes Made:

- We have newly added detailed discussion as **Supplementary Note 5** on page 41 of the revised Supplementary Information.
- We have replaced Supplementary **Fig. 37** with **Fig. R16** on page 42 of the revised Supplementary Information.
- We have newly added the **Figs. R17-R19** as **Supplementary Figs. 38-40** on page 43-45 of the revised Supplementary Information.
- We have revised the discussion on page 15, line 3 of the revised Manuscript.

“The surface structure of (RuIr)O₂ was constructed by considering the oxidation state of Ir and its position within the surface structure. (**Supplementary Figs. 37-40 and Note 5**).”

Comments 2-7: *The authors should carefully check the full text and improve the accuracy of the expression.*

Examples are as follows:

Author’s response: We sincerely apologize for the inaccuracies and any confusion caused by the formatting and expression errors in the manuscript. In response to the reviewer’s requests outlined in comments a) to e), we have conducted a thorough review of the entire text to identify and correct

any errors. These corrections have been addressed in the entire revised Manuscript. We appreciate your constructive feedback, which has significantly improved the quality of our manuscript.

a) “radius = 112 pm, cf. 110 pm for Ni” in Page 5, what does “cf.” mean?

Changes Made:

– We have revised the following sentence on page 5, line 10 of the revised Manuscript:

Original sentence: “radius = 112 pm, cf. 110 pm for Ni”

Revised sentence: “with a radius of 112 pm for Ir, compared to 110 pm for Ni”

b) What does “HFR” mean in Fig. 4e?

Author’s response: Thank you for your insightful question. HFR stands for high-frequency resistance, which can be quantified using Electrochemical Impedance Spectroscopy (EIS) analysis. Specifically, resistance values measured at high frequencies are commonly referred to as HFR and are frequently utilized in various papers (*Nat. Commun.* **2023**, 14, 4592; *J. Electrochem. Soc.* **2022**, 169, 064505; *Energy Environ. Sci.* **2021**, 14, 6338; *Adv. Energy Mater.* **2024**, 14, 2304269). The measured frequency of 10 kHz corresponds to the region representing the ohmic resistance on the Nyquist plot. Ohmic resistance is closely linked to membrane health, serving as a valuable indicator of membrane degradation (*Energy Environ. Sci.* **2023**, 16, 5170; *Adv. Energy Mater.* **2021**, 11, 2101998). The measurements were conducted using Galvanostatic Electrochemical Impedance Spectroscopy (GEIS) analysis, with HFR assessed at 10 kHz every 10 minutes to monitor real-time membrane degradation, including the development of pinholes and other forms of membrane degradation. We have added the full name for HFR in the caption of Fig. 4 in the revised Manuscript.

Changes Made:

– We have newly added the full name for HFR in the caption of **Fig. 4** on page 37 of the revised Manuscript:

Newly added sentence: “The HFR on the right y-axis means high-frequency resistance.”

c) “PXRD patterns of Ru NPs/C and RuO₂ NPs/C” in Supplementary Page 18.

Changes Made:

– We have revised the following sentence on page 18 of the revised Supplementary Information:

Original sentence: “(c) PXRD patterns of Ru NPs/C and RuO₂ NPs/C”

Revised sentence: “(c) PXRD patterns of RuO₂ NPs/C. The gray box indicates the remaining metallic species in RuO₂ NPs/C.”

d) “Ru 3p_{3/2} deconvoluted XPS spectra” and “Area ratio of Ru species (%)” in Supplementary Page 29.

Changes Made:

– We have revised the following sentence on page 28 of the revised Supplementary Information:

Original sentence: “Ru 3p_{3/2} deconvoluted XPS spectra”

Revised sentence: “Ru 3p_{3/2} XPS spectra”

– We have replaced the legend of Supplementary Fig. 26 on page 28 of the revised Supplementary Information:

Original legend: “Area ratio of Ru species (%)”

Revised legend: “Area percentage of Ru species (%)”

e) *The First derivatives of Ru K-edge XANES regions for Ru(acac)₃ in Supplementary Fig. 33 is not clear.*

Author’s response: We apologize for the confusion caused regarding the XANES analysis results of reference samples. We have revised the graph of the first derivatives of Ru(acac)₃ in the Ru K-edge XANES spectra to reflect the reviewer’s comments, ensuring the lines are more clearly visible (**Fig. R20**).

Fig. R20 First derivatives of Ru K-edge XANES regions of control group samples: RuO₂, Ru(acac)₃, Ru foil.

Changes Made:

- We have replaced the **Supplementary Fig. S33b** with **Fig. R20** on page 35 of the revised Supplementary Information.

Reviewer #3

In this work, the active atom Ir was integrated into the RuO₂ matrix at the atomic scale, and the controlled growth of Ru_xIr_yO₂ was achieved at the lattice-expanded Ni₃S₄ (e-Ni₃S₄), which maximally suppressed the participation of lattice oxygen in the OER reaction and synergized the good stability of Ru and the high activity of Ir. The catalysts were also driven at 174 mV for 10 mA cm⁻² and operated stably for 360 h. XAS analysis and DFT calculations further verified the electron transfer of Ir and Ru, thus promoting the catalyst AEM pathway. In this study, it was found that the synergistic effect generated by atomic-level multi-elements can effectively enhance the acidic OER activity, which lays the foundation for efficient green hydrogen production. It is recommended that this paper be accepted with following modifications.

We sincerely thank you for your kind and diligent effort in reviewing our manuscript. We have made every effort to follow your valuable suggestions to improve the overall quality of our work. Additionally, we have worked diligently to explain the correlation between the lattice parameters of the Ni₃S₄-based template and the growth rate of the Ru/Ir atoms. Please see below for our point-by-point responses to your comments.

Comments 1: Please depict the growth process of the RuIr alloy in more detail in Scheme 1.

Author's response: Thank you for your insightful comment. As the reviewer suggested, we have added a small atomic illustration for the growth process of RuIr on both Ni₃S₄ and e-Ni₃S₄ templates, as shown in **Fig. R21**. Additionally, to better illustrate the morphology of the Ni₃S₄ template and the dendritic form of the Ru and Ir atoms grown on it, we have revised the entire Scheme. In the revised Manuscript, we have replaced **Scheme 1** with **Fig. R21**.

Fig. R21 Schematic illustration of Ru/Ir oxide-based nanostructures with different Ru/Ir atomic configurations.

Changes Made:

– We have replaced **Scheme 1** with **Fig. R21** on page 33 of the revised Manuscript.

Comments 2: Please give the correspondence between the lattice parameters of the template and the Ru/Ir growth in a more intuitive way.

Author's response: Thank you for your insightful comment. To provide a more intuitive

correspondence between the lattice parameters of the Ni₃S₄-based template and the Ru/Ir growth, we summarized the relevant analysis results and detailed discussion below.

In this study, we introduced larger-sized Ir atoms into the Ni₃S₄ framework to induce partial changes in the lattice structure of Ni₃S₄, aiming to control the growth rate of Ru/Ir atoms and the Ru/Ir atomic configuration within the RuIr shell. The PXRD pattern of e-Ni₃S₄ (**Fig. R22a**) demonstrated a slight shift to a lower angle compared to Ni₃S₄. This shift clearly indicates the lattice expansion due to the embedding of Ir single atoms in the Ni₃S₄ matrix, providing compelling evidence for the structural modification achieved through Ir doping. This approximately 6% lattice expansion was further validated by the d-spacings of Ni₃S₄ facets obtained from HAADF-STEM images and their corresponding FFT patterns (**Supplementary Fig. 2**). The detailed lattice changes for each facet in the Ni₃S₄ phase of the Ni₃S₄ and e-Ni₃S₄ were summarized in **Figure R22b** and **Table R5**.

Fig. R22 a PXRD patterns of Ni₃S₄ and e-Ni₃S₄. **b** Comparison of the d-spacing value of each facet of Ni₃S₄ phase in Ni₃S₄ (blue circle) and e-Ni₃S₄ (red square).

Facet	Measured d-spacing	Measured d-spacing	d-spacing values (reference)
	of Ni ₃ S ₄	of e-Ni ₃ S ₄	
(2̄2̄2̄)	0.165 nm	0.176 nm	0.167 nm
(2̄2̄2̄)	0.235 nm	0.255 nm	0.236 nm
(2̄00)	0.272 nm	0.283 nm	0.272 nm
(02̄2̄)	0.273 nm	0.284 nm	0.272 nm

$(\bar{3}1\bar{1})$	0.197 nm	0.216 nm	0.196 nm
(400)	0.135 nm	0.146 nm	0.136 nm

Table R5. Reference potential ranges for the oxidation peaks of Ru and Ir species in cyclic voltammetry (CV) curves.

The expansion of the lattice parameter in the template is crucial for facilitating the incorporation of Ru and Ir atoms during the growth process. As shown in **Fig. R23**, the growth behavior of Ru/Ir depends on the lattice parameter of the template. When the pristine Ni_3S_4 was reacted simultaneously with Ru^{3+} and Ir^{3+} via solvothermal synthesis, rapid Ru deposition followed by Ir deposition on the Ni_3S_4 was observed during the early stages of the reaction (**Figure R23a-d**), resulting in the formation of a Ru@Ir inner shell@outer shell configuration (**Figure R23e**). However, for the e- Ni_3S_4 , the lattice expansion caused by Ir doping increased the lattice mismatch between Ru and the template, reducing the initial rate of Ru deposition (**Figure R23f-i**). This promotes atomic-level mixing of Ru and Ir, finally facilitating more uniform and stable growth of the RuIr alloy (**Figure R23j**).

Fig. R23 Schematic illustration of Ru and Ir growth on **a** Ni₃S₄ and **f** lattice-expanded- Ni₃S₄ (e-Ni₃S₄), which afforded Ru@Ir and RuIr shell configuration, respectively. Comparison of Ru/Ir atomic composition of **b** Ni₃S₄ and **g** e-Ni₃S₄ depending on different reaction times, obtained by EDS analysis. **c, h** PXRD patterns and **d, i** HAADF-STEM images (scale bar = 10 nm) of Ni₃S₄@RuIr and e-Ni₃S₄@RuIr obtained at reaction times of 5-, 10-, and 30-min. Color bars and asterisks in PXRD patterns indicate the reference peaks of *hcp* Ru (yellow, #01-088-2333), *fcc* Ir (purple, #06-0598), and Ni₃S₄ (green, #01-076-1813). EDS line profile analysis of **e** Ni₃S₄@RuIr and **j** e-Ni₃S₄@RuIr determined along the lines marked by arrows in **d** and **i**, respectively.

DFT calculations further examined the correlation between the lattice size of the template and the growth behavior of Ru/Ir atoms (**Fig. R24**). We hypothesized that if the average interatomic bond lengths (d_{ave}) value at the interface between the template (core, Ni₃S₄, and e-Ni₃S₄) and the growing metal (shell, Ru only, and RuIr alloy) matches the d_{ave} value of the standard bulk (Ru_{Bulk} and RuIr_{Bulk}), then the growth of Ru and RuIr shell on each template would be preferred. When Ru atoms were grown on the templates, the difference in d_{ave} between Ru bulk and e-Ni₃S₄@Ru (0.233 Å) was notably larger than that between bulk Ru and Ni₃S₄@Ru (0.098 Å). This finding suggested an unfavorable lattice spacing between the Ru atoms and the e-Ni₃S₄ surface, indicating a significant lattice mismatch. Conversely, the difference in d_{ave} between RuIr bulk and e-Ni₃S₄@Ru (0.121 Å) was similar to that between RuIr bulk and Ni₃S₄@Ru (0.144 Å), which suggested that the stability of the RuIr shell was maintained on the expanded core. Therefore, our experimental results that template lattice modulation plays a pivotal role in determining the Ru/Ir growth behavior were confirmed by theoretical simulation.

Fig. R24 DFT calculations for interatomic bond length (d_{ave}). **a** The optimized bulk structures (Ru_{Bulk}, RuIr_{Bulk}) and shell structures (Ru on Ni₃S₄, RuIr on Ni₃S₄, Ru on e-Ni₃S₄, RuIr on e-Ni₃S₄). The shell structures were extracted from the optimized M (M = Ru, RuIr) on (e-)Ni₃S₄ structures. **b** The averaged interatomic bond lengths of metal and alloy shells on Ni₃S₄ and e-Ni₃S₄ cores.

In summary, the lattice expansion in the e-Ni₃S₄ template is crucial for facilitating the growth and stability of the RuIr alloy. This key insight is strongly supported by both our experimental and computational results. This explanation is detailed under the ‘Preparation of mixed rutile-type (RuIr)O₂/C’ section in the revised Manuscript.

Comments 3: XRD comparisons of Ni₃S₄ and e-Ni₃S₄ should be provided.

Author’s response: We appreciate the reviewer’s constructive comment on our manuscript. As the reviewer pointed out, PXRD analysis is crucial for confirming the lattice expansion of Ni₃S₄ induced by the introduction of Ir into the Ni₃S₄ matrix. As shown in **Fig. R25a**, the PXRD pattern of e-Ni₃S₄ showed a slight shift to lower angles compared to that of Ni₃S₄, indicating the lattice expansion due to the embedding of Ir atoms in the Ni₃S₄ matrix, which provides compelling evidence for the structural modification achieved through Ir doping. Moreover, the introduction of larger-sized Ir atoms (with a radius of 112 pm for Ir, compared to 110 pm for Ni) into the Ni₃S₄ framework led to an approximately 6% lattice expansion, as confirmed by the d-spacings of Ni₃S₄ facets extracted from HAADF-STEM and corresponding FFT patterns (**Fig. R25b**).

We have revised the **Fig. 1** and the discussion to reflect the importance of the observed lattice structure changes, as demonstrated by PXRD and d-spacing analysis in the revised Manuscript. Additionally, the PXRD analysis results measured in the range of $20^\circ - 60^\circ$, and the differences in d-spacing for more facets are summarized in **Supplementary Fig. 2** in the revised Supplementary Information.

Fig. R25 Growth mechanism of Ru/Ir on two distinct templates. **a** PXRD patterns of Ni_3S_4 and $\text{e-Ni}_3\text{S}_4$. **b** Comparison of the d-spacing value of (311) and (400) facets of the Ni_3S_4 phase, observed in PXRD patterns in Fig. 1a. The two dashed lines represent the reference d-spacing values for the (311) and (400) facets. Schematic illustration of Ru and Ir growth on **c** Ni_3S_4 and **d** lattice-expanded- Ni_3S_4 ($\text{e-Ni}_3\text{S}_4$), which afforded Ru@Ir and RuIr shell configurations, respectively. Green and red spheres denote Ni and S atoms, respectively. **e, h** PXRD patterns and **f, i** HAADF-STEM images (scale bar = 10 nm) of Ni_3S_4 @RuIr and $\text{e-Ni}_3\text{S}_4$ @RuIr obtained at reaction times of 5-, 10-, and 30-min. Color bars and asterisks in PXRD patterns

indicate the reference peaks of *hcp* Ru (yellow, #01-088-2333), *fcc* Ir (purple, #06-0598), and Ni₃S₄ (green, #01-076-1813). EDS line profile analysis of **g** Ni₃S₄@RuIr and **j** e-Ni₃S₄@RuIr determined along the lines marked by arrows in **f** and **i**, respectively. **k** Final images of the AIMD trajectories of Ni₃S₄@RuIr and e-Ni₃S₄@RuIr. Light green, yellow, turquoise, and navy spheres denote Ni, S, Ru, and Ir, respectively. **l** Energy profiles for the AIMD trajectories in **k**, with the gray region indicating the 10 ps equilibrium process.

Changes Made:

- We have replaced **Fig. 1** with **Fig. R25** on page 34 of the revised Manuscript.
- We have revised the discussion for the lattice structure of two types of Ni₃S₄-based templates on page 5, line 11 of the revised Manuscript.

“Notably, the powder X-ray diffraction (PXRD) pattern of e-Ni₃S₄ (**Fig. 1a** and **Supplementary Fig. 2e**) demonstrated a slight shift to a lower angle compared to Ni₃S₄, indicating the lattice expansion due to the embedding of Ir single atoms in the Ni₃S₄ matrix. The d-spacing analysis of Ni₃S₄ facets, derived from high-angle annular dark-field scanning transmission electron microscopy (HAADF-STEM) images, and their corresponding fast Fourier transform (FFT) patterns (**Fig. 1b** and **Supplementary Fig. 2d,f**) revealed approximately 6% lattice expansion in the Ni₃S₄ phase of e-Ni₃S₄, providing compelling evidence of the structural modification achieved through Ir doping.”

Comments 4: *Please explain what does the arrows mean in Figure 2(b).*

Author’s response: Thank you for your detailed comment. The arrows depicted in **Fig. 2b** indicated the range of the line profile, as shown in **Fig. 2c**. In the original version of the manuscript, the caption for **Fig. 2c** stated, ‘EDS line profile analysis of (RuIr)O₂/C for the marked area in panel (b).’ However, to reduce confusion and reflect the reviewer’s comments, we have made detailed modifications to the caption in the revised Manuscript.

Changes Made:

- We have revised the following caption of **Fig. 2** on page 35 of the revised Manuscript.

Revised sentences: “**b** Combined and individual EDS elemental mapping images of Ru (yellow), Ir (purple), and O (cyan). The white arrow indicates the range of the line profile in Fig. 2c. **c** Line profile analysis of (RuIr)₂O₂/C corresponding to the marked area indicated by the white arrow in Fig. 2b.”

Comments 5: Please label the fitted line for the Tafel slope in Figure 3(b).

Author’s response: Thanks for the constructive comments on electrocatalytic measurements. In response to the reviewer’s comments, we have added a fitted line in the form of a dotted line to the Tafel plot in **Fig. 3b** of the revised Manuscript to determine the Tafel slope.

Changes Made:

- We have replaced **Fig. 3b** with **Fig. R26** on page 36 of the revised manuscript.
- We have revised the following caption of **Fig. 3b** on page 36 of the revised manuscript:

Original caption: “**b** Tafel plots constructed based on the curves in (a)”

Revised caption: “**b** Tafel plots constructed based on the curves in Fig. 3a with fitted lines for Tafel slope.”

Fig. R26 Tafel plots with fitted lines for Tafel slope.

Comments 6: *It should be indicated whether the dissolution of Ru and Ir elements occurred after the stability test.*

Author's response: We appreciate the opportunity to address the points you raised and to clarify our findings. To examine the structural changes and dissolution properties of Ru and Ir elements in Ru/Ir oxide-based electrocatalysts during OER operation, we conducted various structural characterizations.

First, TEM images (**Supplementary Fig. 21**) and EDS elemental mapping (**Supplementary Fig. 22**) revealed that the dendritic shell thickness of (RuIr)O₂/C remained unchanged, with no noticeable detachment of the Ru/Ir shell, suggesting preserved structural integrity without dissolution. In contrast, RuO₂@IrO₂/C showed significant thinning of the dendritic shell due to particle detachment (**Supplementary Fig. 23**). In addition, HRTEM images and corresponding FFT patterns (**Supplementary Fig. 24**) indicated that (RuIr)O₂/C maintained its crystallinity, whereas RuO₂@IrO₂/C experienced deterioration due to Ru and Ir leaching. Finally, the ICP-MS analysis confirmed negligible leaching of Ru and Ir from (RuIr)O₂/C, while RuO₂@IrO₂/C exhibited considerable loss of these elements during long-term OER operation (**Supplementary Fig. 25**). In summary, the Ru and Ir atoms in (RuIr)O₂/C remain robust and maintain structural integrity during OER operation due to the synergistic interaction between Ru and Ir on an atomic scale, as demonstrated by in-situ XAFS (**Figs. 5, 6**) and XPS (**Supplementary Figs. 26, 27**) analyses in this manuscript. In contrast, the Ru and Ir atoms in RuO₂@IrO₂/C significantly leach out, resulting in a highly unstable structure.

To further validate the stability of Ru and Ir elements in (RuIr)O₂/C against dissolution, we calculated dissolution potentials (U_{diss}) using the DFT calculations with the following equation (*Adv. Funct. Mater.* **2024**, 2401095):

$$U_{diss} = U_0 + \frac{1}{ne} \{E(M_{m-1}O_{2m}) + \mu_M - E(M_mO_{2m})\}$$

where n and m correspond to the number of electrons involved in the reduction reaction of metal ions and the number of metal atoms in the system, respectively. The U_0 , E , and μ_M represent the standard reduction potential of the metal, the electronic energy of the system, and the chemical potential of the metal, respectively. μ_M was calculated as the DFT energy per atom of bulk Ru and Ir. The U_0 value is known to be 0.46 V for Ru ($\text{Ru}^{2+} + 2e^- \rightarrow \text{Ru}$) and 1.16 V for Ir ($\text{Ir}^{3+} + 3e^- \rightarrow$

Ir) (*Electrochim. Acta.* **2007**, 52, 5829). For the (RuIr)O₂ surface, we considered the surface structures with a probability greater than 0.05 (Supplementary Fig. 43). The calculated U_{diss} values are shown in Fig. R27. The U_{diss} values of the metal elements (Ru, Ir) in the M_{6c} site and AS positions indicated that the Ru atom in the M_{6c} site of (RuIr)O₂ had a U_{diss} value 0.16 V higher than that of Ru in the M_{6c} site of RuO₂. Although the Ir atom in the AS of (RuIr)O₂ had a slightly lower U_{diss} value (1.53 V) than the Ir atom in the AS position of IrO₂ (1.60 V), this value is still higher than the U_{diss} of the AS atom in RuO₂. Therefore, we conclude that incorporating Ir into RuO₂ significantly enhances the dissolution stability of the (RuIr)O₂ catalyst surface.

Fig. R27 The DFT calculated U_{diss} on RuO₂, IrO₂ and (RuIr)O₂. For the values of (RuIr)O₂, the standard deviation is indicated in parentheses.

Comments 7: Please elaborate on how Ru, Ir synergistically and effectively inhibits the LOM mechanism and promotes the AEM process.

Author's response: Thanks for the reviewer's insightful comment on OER mechanisms. In response, we have further examined the lattice oxygen mechanism (LOM) and evaluated its catalytic activity for RuO₂, IrO₂, and (RuIr)O₂. For the (RuIr)O₂ surface, we considered the surface structures with a probability greater than 0.05 (Supplementary Fig. 43) and averaged their LOM overpotentials. The following step reactions were considered as LOM (*ACS Catal.* **2020**, 10, 3650):

where M-O and M-(vac) correspond to the lattice oxygen atom and lattice oxygen vacancy, respectively, which directly participates in the LOM. The adsorption Gibbs free energies were calculated as:

$$\Delta G_{\text{OH}^*} = E(\text{OH}^*) + 0.5 \cdot \mu(\text{H}_2) - E(*) - \mu(\text{H}_2\text{O}) + G_{\text{corr, OH}^*}$$

$$\Delta G_{\text{O}^*} = E(\text{O}^*) + \mu(\text{H}_2) - E(*) - \mu(\text{H}_2\text{O}) + G_{\text{corr, O}^*}$$

$$\Delta G_{\text{M-OH}} = E(\text{M-OH}) + 1.5 \cdot \mu(\text{H}_2) + \mu(\text{O}_2) - E(*) - 2 \cdot \mu(\text{H}_2\text{O}) + G_{\text{corr, M-OH}}$$

While we observed higher OER activity for (RuIr)O₂ compared to RuO₂ and IrO₂ for LOM (**Fig. R28b**), the calculated overpotentials are substantially higher than those of AEM (**Fig. R28a**). This result is consistent with previous theoretical (*ACS Catal.* **2020**, 10, 3650; *Chem* **2023**, 9, 3600) and experimental (*Energy Environ. Sci.* **2022**, 15, 1988) findings, which suggest that the contribution of lattice oxygens to the overall OER activity is negligible for Ru-based and Ir-based catalysts. Therefore, these results indicate that (RuIr)O₂ exhibits superior OER activity compared to RuO₂ and IrO₂, particularly highlighting the efficiency of the OER process via the AEM pathway.

Fig. R28 The Gibbs free energy diagram of **a** AEM and **b** LOM on RuO₂, IrO₂ and (RuIr)O₂. For the values of (RuIr)O₂, the standard deviation is indicated in parentheses. The red and white balls and yellow and purple polyhedrons in the atomic structures correspond to O, H, Ru, and Ir atoms.

Additionally, as the reviewer commented, we further carried out in-situ differential electrochemical mass spectrometry (DEMS) analyses using the isotope ¹⁸O to verify the suppressed lattice oxygen participation on (RuIr)O₂ catalysts during the OER (*Nat. Commun.* **2024**, 15, 2501; *J. Am. Chem. Soc.* **2021**, 143, 6482). Before DEMS measurement, the ¹⁸O-labelled (RuIr)O₂ and homemade-RuO₂ catalysts were

prepared by CV cycling in a 0.1 M HClO₄ in heavy-oxygen water (H₂¹⁸O). Then, the evolved O₂ was measured by DEMS in a 0.1 M HClO₄ electrolyte of H₂¹⁶O (**Fig. R29**). The signals of the ³⁴O₂ indicate the direct ¹⁶O-¹⁸O coupling, where the ¹⁶O originates from water and ¹⁸O originates from the lattice oxygen (*Nat. Chem.* **2017**, 9, 457). The participation ratio of lattice oxygen (LOM%) was evaluated by the ratio of ³⁴O₂ to (³²O₂ + ³⁴O₂). As shown in **Fig. R30**, the LOM% of the (RuIr)O₂ was only 0.373%, whereas the LOM% of the homemade-RuO₂ (3.521%) was about ~9.4-fold higher than that of (RuIr)O₂. Therefore, the lattice oxygen participation during the OER was significantly hindered in the (RuIr)O₂, which corroborates with its high OER stability over RuO₂ under acidic conditions.

To provide a detailed explanation of the OER mechanism of (RuIr)O₂/C, demonstrated by DFT calculations and DEMS measurements, we have added a new section titled ‘Enhanced OER mechanism of (RuIr)O₂/C via AEM pathway’ in the revised Manuscript.

Fig. R29 DEMS measurements for ³⁴O₂ (¹⁶O¹⁸O, yellow) and ³²O₂ (¹⁶O¹⁶O, green) of **a, c** (RuIr)O₂/C and **b, d** homemade-RuO₂/C.

Fig. R30 DEMS signals of $^{34}\text{O}_2$ ($^{16}\text{O}^{18}\text{O}$) and $^{32}\text{O}_2$ ($^{16}\text{O}^{16}\text{O}$) from the evolved O_2 for the ^{18}O -labelled **a** $(\text{RuIr})\text{O}_2/\text{C}$ and **b** homemade- RuO_2/C in 0.1 M HClO_4 solution in H_2^{16}O . **c** Percentage contribution of lattice oxygen (LOM%) in the OER.

Changes Made:

- We have newly added **Fig. R28b** as **Supplementary Fig. 44** on page 49 of the revised Supplementary Information.
- We have newly added **Supplementary Note 6** to present a detailed calculation method using the LOM pathway on page 50 of the revised Supplementary Information.
- We have newly added **Fig. R29** as **Supplementary Fig. 47** on page 53 of the revised Supplementary Information.
- We have newly added **Fig. R30** as **Fig. 7f,g** on page 40 of the revised Manuscript.
- We have revised the following discussion to explain the DFT calculations for the LOM pathway on page 15, line 11 of the revised Manuscript.

“Utilizing these surface structures, we calculated the Gibbs free energy changes of OER intermediates for all sites across 38 structures and 3 sites (114 calculations in total), following the AEM. The corresponding weighted averages were then determined using the Boltzmann probability, with more stable surface structures contributing more significantly to material properties (**Supplementary Fig. 43**). During the AEM pathway (**Fig. 7b,c**), the Ir active sites in $(\text{RuIr})\text{O}_2$ exhibited weaker affinities for all oxygen intermediates (OH^* , O^* , OOH^*) compared to those in IrO_2 . Consequently, $(\text{RuIr})\text{O}_2$ demonstrated a higher OER activity

(overpotential (η) = 0.48 V) than pure rutile-type oxides (η = 0.76 V for IrO₂, η = 0.62 V for RuO₂). Although (RuIr)O₂ exhibited better OER activity than RuO₂ and IrO₂ through the LOM pathway (**Supplementary Fig. 44** and **Supplementary Note 6**), the calculated overpotentials were significantly higher compared to those of the AEM pathway. Therefore, (RuIr)O₂ exhibits superior OER activity compared to RuO₂ and IrO₂, particularly highlighting the efficiency of the OER process via the AEM pathway.”

- We have newly added the detailed discussion for DEMS analysis on page 16, line 18 of the revised Manuscript.

“We further carried out in-situ differential electrochemical mass spectrometry (DEMS) analyses using the isotope ¹⁸O to investigate to verify the suppressed lattice oxygen participation on (RuIr)O₂ catalysts during the OER^{66,67}. Before DEMS measurement, the ¹⁸O-labelled (RuIr)O₂ and home-made RuO₂ catalysts were prepared by CV cycling in a 0.1 M HClO₄ in heavy-oxygen water (H₂¹⁸O). Then, the evolved O₂ was measured by DEMS in a 0.1 M HClO₄ electrolyte of H₂¹⁶O (**Supplementary Fig. 47**). The signals of the ³⁴O₂ indicate the direct ¹⁶O-¹⁸O coupling, where the ¹⁶O originates from water and ¹⁸O originates from the lattice oxygen⁶⁸. The participation ratio of lattice oxygen (LOM%) was evaluated by the ratio of ³⁴O₂ to (³²O₂ + ³⁴O₂). As shown in **Fig. 7f, g**, the LOM% of the (RuIr)O₂ was only 0.373%, whereas the LOM% of the homemade-RuO₂ (3.521%) was about ~9.4-fold higher than that of (RuIr)O₂. Therefore, the lattice oxygen participation during the OER was significantly hindered in the (RuIr)O₂, which corroborates with its high OER stability over RuO₂ under acidic conditions.”

- We have newly added the DEMS measurement method on page 23, line 21 of the revised Manuscript.

“In-situ DEMS involving H₂¹⁸O was carried out by an HPR-40 quadrupole mass spectrometer system (HIDEN Analytical Limited, England) and Type A cell to identify the level of participation of lattice oxygen mechanism during the OER⁶⁵. The catalysts were drop-casted on polished GC electrodes (5 mm in diameter) with a catalyst loading of 40 $\mu\text{g cm}^{-2}$. The catalyst-loaded GC electrode, Ag/AgCl electrode, and Pt wire were used as the working, reference, and counter electrode, respectively. The ¹⁸O isotope labeling of the catalysts was conducted by 5 CV cycles at a scan rate of 5 mV s⁻¹ in 0.1 M HClO₄ electrolyte containing

H₂¹⁸O. During the labeling, the electrolyte was pumped through the cell at a flow rate of 0.9 mL s⁻¹. The potential range of CV cycles was set at 1.25–1.65 V_{RHE} for RuIrO₂ and 1.25–1.95 V_{RHE} for RuO₂, considering a similar current density. Then, the resulting electrodes were rinsed with H₂¹⁶O several times to remove residual H₂¹⁸O. Finally, the ¹⁸O-labelled electrodes were placed in 0.1 M HClO₄ containing H₂¹⁶O, and CV was carried out within the same potential windows. During the CV, mass spectrometry was used to detect ³²O₂ (¹⁶O¹⁶O) and ³⁴O₂ (¹⁶O¹⁸O) generated during the OER process. The mass spectroscopy signals were baseline subtracted.”

Comments 8: *Whether the cell voltage was iR-compensated and whether it was necessary.*

Author’s response: Thank you for your insightful question. Generally, iR compensation is employed to assess the intrinsic activity of the catalyst itself, excluding the solution resistance, in half-cell studies. In unit cell tests, the “R” value in iR-compensated refers to ohmic resistance, representing the membrane resistance of the Membrane Electrode Assembly (MEA). Since the membrane is a critical component of the MEA, excluding this part and evaluating the performance of the unit cell may not truly reflect the activity of the unit cell. However, considering that membrane thickness, type, and other factors can significantly impact performance, this metric can be useful when comparing different types or thicknesses of membranes. The Severin Vierrath group noted differences in activity between commercial and developmental membranes after measuring activity and Electrochemical Impedance Spectroscopy (EIS), followed by Internal Resistance (IR) correction to account for membrane variations (*Adv. Energy Mater.* **2020**, 1903995). Therefore, the decision to apply IR correction or not will depend on the specific analysis being conducted (*Adv. Energy Mater.* **2021**, 11, 2101998).

In this manuscript, the PEMWE measurement data presented in Figure 4 are not iR-compensated. However, to illustrate the comparison of iR compensation before and after for (RuIr)O₂/C and commercial IrO₂ electrocatalysts, we have added the polarization curves for single-cell performance in PEMWE (**Fig. R31**).

Fig. R31 Polarization curves for single-cell performance in PEMWE of **a** (RuIr)O₂/C and **b** commercial IrO₂ electrocatalysts with and without iR compensation.

Point-by-point responses to reviewers' comments.

Reviewer #1

The revisions and responses to the review comments are satisfied.

Thank you.

Reviewer #2

In the revised manuscript of NCOMMS-24-20531A, the authors have added some experimental and calculational results (XPS, HRTEM, and Bader charge analysis) to support the structure and performance of (RuIr)O₂/C electrocatalyst according to the reviewers' suggestions. The quality of the manuscript has been improved, and now I recommend the publication of this work in Nature Communications. However, the authors should address the following issues.

We sincerely thank you for your great effort in reviewing our manuscript. We also appreciate your careful reading, enlightening comments, and valuable suggestions, which have significantly helped improve our manuscript. We have made every effort to address the comments thoroughly, and our specific replies and modifications are listed as follows:

Comments 1: *The authors claimed that a large fraction of Ni₃S₄ maintained its presence at OER potentials. The reason for the difference in S 2p XPS spectra before and after the 24 h OER operation should be explained (Supplementary Fig. 15 and Fig. R7b).*

Author's response: Thank you for your insightful comment. The new peak observed in the S 2p XPS spectra after the OER operation, marked in gray section (Fig. R1b), suggests that a small fraction of sulfur atoms in the Ni₃S₄ template may have been oxidized to sulfate species (SO₄²⁻) under the high potential conditions. However, this oxidation appears to be limited to the surface layer of Ni₃S₄ and does not imply a complete transformation or compromise its overall structural integrity.

Importantly, the core structure of Ni₃S₄ remains largely intact and stable during the OER, consistent with our claim of its durability under these conditions. The surface-confined oxidation does not significantly affect the crystal or electronic structure of the Ni₃S₄. Therefore, the changes observed in the S 2p XPS spectra of (RuIr)O₂/C after OER operation reflect the oxidation of only a small number of surface sulfur atoms, which does not substantially impact the stability of the Ni₃S₄ template.

Fig. R1 S 2p XPS spectra of (RuIr)O₂/C and RuO₂@IrO₂/C **a** before and **b** after OER operation. The gray section in Fig. R1b indicates the presence of sulfate species (SO₄²⁻) after OER operation.

Comments 2: *The authors were encouraged to add experimental results of SiO₂ as a sacrificial substrate into the Supplementary Information to better illustrate the role of carbon substrate.*

Author's response: We appreciate the reviewer's constructive comment on our manuscript. As shown in Fig. R2a, we synthesized unsupported (RuIr)O₂ and RuO₂@IrO₂ electrocatalysts using SiO₂ as a sacrificial substrate instead of carbon. This approach allowed us to eliminate the potential impact of carbon corrosion during the OER operation and assess the effects of Ru/Ir atomic configurations on the OER performance of both catalysts.

From the additional experimental analyses (Figs. R2-R4), we concluded that while the carbon substrate enhances electrochemical performance by preventing nanoparticle aggregation and improving electrical conductivity, the Ru-Ir atomic interactions within the mixed rutile-type oxide phase are the primary factor driving the high OER performance. In line with the reviewer's suggestion, we have included the experimental results and detailed discussions regarding the use of SiO₂ as a sacrificial substrate in the revised Supplementary Information and Manuscript to demonstrate the role of the carbon substrate.

Fig. R2 Structural characterization of unsupported Ru/Ir oxide-based electrocatalysts. a Schematic illustration and **b** TEM images for the formation of unsupported $\text{RuO}_2@\text{IrO}_2$ and $(\text{RuIr})\text{O}_2$. **c** PXRD patterns for unsupported $\text{RuO}_2@\text{IrO}_2$ and $(\text{RuIr})\text{O}_2$. Gray boxes denote the remaining metallic species in unsupported $\text{RuO}_2@\text{IrO}_2$ and $(\text{RuIr})\text{O}_2$, which show the same behavior as with carbon support.

Fig. R3 Atomic composition analysis of unsupported $\text{RuO}_2@ \text{IrO}_2$ and $(\text{RuIr})\text{O}_2$. **a** Combined and individual EDS elemental mapping images of O (cyan), Ru (yellow), and Ir (purple) within unsupported **a** $\text{RuO}_2@ \text{IrO}_2$ and **b** $(\text{RuIr})\text{O}_2$. Line profile analysis for **c-e** $\text{RuO}_2@ \text{IrO}_2$ and **f-h** $(\text{RuIr})\text{O}_2$ corresponding to the marked area indicated by the white arrow in panel **a** and **b**, respectively.

Fig. R4 OER activity of unsupported (RuIr)O₂ and RuO₂@IrO₂. **a** OER polarization curves of unsupported (RuIr)O₂ and RuO₂@IrO₂. **b** Overpotential of unsupported (RuIr)O₂ and RuO₂@IrO₂ to drive 10 mA cm⁻² of current density.

Changes Made:

- We have newly added the Figs. R2-R4 as Supplementary Figs. 21-23 on page 24-26 of the revised Supplementary Information.
- We have newly added detailed discussion as Supplementary Note 4 on page 23 of the revised Supplementary Information.
- We have newly added detailed discussion on page 9, line 17 of the revised Manuscript.

Reviewer #3

In this research, the integration of active Ir atoms into the RuO₂ matrix by a controlled method can greatly optimize the synergistic OER between Ru and Ir activity centers. As a results, the optimized (RuIr)O₂/C exhibits an excellent overpotential of only 174 mV at 10 mA/cm² and a good stability of 100 mA/cm². In further application to PEMWE, it also demonstrates its ability to operate stably at 1 A/cm². In situ characterization and DFT calculations further illustrate the significant role of the atomic level mixing of Ru and Ir in optimizing the adsorption energy. It is recommended that the manuscript be considered for acceptance with the following revisions.

We sincerely thank you for your thoughtful and diligent efforts in reviewing our manuscript. We have carefully considered your valuable suggestions and have made every effort to improve the overall quality of our work. Please find below our point-by-point responses to your comments.

Comments 1: *The keywords in the manuscript does not highlight the key points and highlights of the paper, please rethink and revise it.*

Author's response: Thank you for your helpful comments on our manuscript. We appreciate the opportunity further to clarify the novelty and key breakthroughs of our study. Our study provides a comprehensive understanding of the Ru-Ir alloy system, particularly the mixed rutile-type Ru/Ir oxide phase, and elucidates the mechanisms contributing to its high OER activity and stability. Notably, we introduce a novel synthesis strategy that leverages template lattice parameter modulation to optimize the atomic configuration and growth of the Ru/Ir atoms.

To better highlight the key points of this manuscript, we have revised the keywords as follows: oxygen evolution reaction (OER), proton exchange membrane water electrolysis (PEMWE), lattice structure modulation, atomic-level mixing, and mixed rutile oxide. Additionally, we have also revised the Table of Contents (TOC) figure to provide a clear and concise presentation of the key novelty and concept of this study.

Fig. R5 Table of Contents.

Changes Made:

- We have replaced the TOC figure with Fig. R5 on page 42 of the revised Manuscript.
- We have revised the TOC on page 42 of the revised Manuscript to highlight the novelty of this study.

Comments 2: *Please describe the unique advantages and rationale for the selection of Ni₃S₄ and e-Ni₃S₄ as templates for atomic level mixing.*

Author's response: Thank you for your insightful comment. We appreciate the opportunity to clarify the novelty and breakthroughs of our study. In this study, we precisely controlled the growth rate, degree of mixing, and crystal growth direction of Ru and Ir atoms to develop a mixed rutile-type oxide phase ((RuIr)O₂) by utilizing the lattice parameter modulation of the templates.

1. **Control over the lattice variation:** In metal compounds, the strategic arrangement of cations and anions forms an ideal framework that facilitates the accommodation of foreign atoms, thereby inducing lattice modifications. The spinel structure of Ni₃S₄ is particularly known for its high lattice flexibility, which allows the stable incorporation of foreign atoms at an atomic scale while preserving the overall structural integrity. This makes Ni₃S₄ an ideal platform for achieving precise atomic-level mixing, crucial for enhancing catalytic performance.
2. **Achieving lattice expansion:** We introduced larger-sized Ir atoms (with a radius of 112 pm for Ir, compared to 110 pm for Ni) into the Ni₃S₄ framework, inducing lattice modifications and successfully forming lattice-expanded Ni₃S₄ (e-Ni₃S₄) (Fig. R6). This expansion amplified the lattice mismatch between Ru and e-Ni₃S₄ surface, effectively slowing the initial deposition of Ru atoms (Fig. R7). This deceleration promoted the co-deposition of Ru and Ir, enabling a more uniform and stable growth of the RuIr alloy. The atomic-scale mixing of Ru and Ir atoms enhances their interaction across active sites, leading to significant improvements in both the catalytic activity and stability during the acidic OER process.
3. **Stabilization of the mixed alloy:** The lattice mismatch between Ru and the e-Ni₃S₄ surface creates a kinetically favorable environment for RuIr alloy formation compared to the pristine Ni₃S₄ surface (Fig. R8). This enhanced environment fosters the stabilization of the RuIr alloy shell, a critical factor in boosting the long-term durability of the catalyst, particularly under high current densities during OER operation.

Therefore, the surface lattice parameters of Ni₃S₄ offer a versatile and tunable platform for the incorporation of Ir atomic dopants and precise modulation of Ru and Ir deposition rates. This meticulous control over atomic-level mixing is crucial for establishing the catalytic synergy required to achieve superior OER performance.

Fig. R6 **a** XRD patterns of Ni_3S_4 and $e\text{-Ni}_3\text{S}_4$. **b** Comparison of the d-spacing value of each facet of Ni_3S_4 phase in Ni_3S_4 (blue circle) and $e\text{-Ni}_3\text{S}_4$ (red square).

Fig. R7 Schematic illustration of Ru and Ir growth on **a** Ni_3S_4 and **f** lattice-expanded- Ni_3S_4 ($e\text{-Ni}_3\text{S}_4$), which afforded Ru@Ir and RuIr shell configuration, respectively. Comparison of Ru/Ir atomic composition of **b** Ni_3S_4 and **g** $e\text{-Ni}_3\text{S}_4$ depending on different reaction times, obtained by EDS analysis. **c**, **h** XRD patterns and **d**, **i** HAADF-STEM images (scale bar = 10 nm) of Ni_3S_4 @RuIr and $e\text{-Ni}_3\text{S}_4$ @RuIr obtained at reaction times of 5-, 10-, and 30-min. Color bars and

asterisks in PXRD patterns indicate the reference peaks of *hcp* Ru (yellow, #01-088-2333), *fcc* Ir (purple, #06-0598), and Ni_3S_4 (green, #01-076-1813). EDS line profile analysis of **e** $\text{Ni}_3\text{S}_4@\text{RuIr}$ and **j** $\text{e-Ni}_3\text{S}_4@\text{RuIr}$ determined along the lines marked by arrows in **d** and **i**, respectively.

Fig. R8 DFT calculations for interatomic bond length (d_{ave}). **a** The optimized bulk structures (Ru_{Bulk} , $\text{RuIr}_{\text{Bulk}}$) and shell structures (Ru on Ni_3S_4 , RuIr on Ni_3S_4 , Ru on $\text{e-Ni}_3\text{S}_4$, RuIr on $\text{e-Ni}_3\text{S}_4$). The shell structures were extracted from the optimized M (M = Ru, RuIr) on (e-) Ni_3S_4 structures. **b** The averaged interatomic bond lengths of metal and alloy shells on Ni_3S_4 and $\text{e-Ni}_3\text{S}_4$ cores.

Comments 3: Please give the performance comparison of (RuIr)O₂/C at different current densities (10, 100 mA cm⁻²) in the LSV plot of Fig. 3.

Author's response: We appreciate the reviewer's helpful comments. As the reviewer suggested, we have compared the overpotentials required to drive current densities of 10 mA cm⁻² and 100 mA cm⁻² for (RuIr)O₂/C and other control-group catalysts, which was derived from the LSV graph in Fig. 3a.

As shown in the graph (Fig. R9), (RuIr)O₂/C demonstrated significantly lower overpotentials at both current densities, highlighting its superior catalytic performance compared to RuO₂@IrO₂/C and RuO₂ NPs/C. The overpotentials of 174 mV at 10 mA cm⁻² and 246 mV at 100 mA cm⁻² for (RuIr)O₂/C underscore the effectiveness of the Ru-Ir atomic interaction within the mixed rutile-type oxide phase, contributing to its enhanced OER performance. In contrast, RuO₂@IrO₂/C and RuO₂ NPs/C exhibited notably higher overpotentials, particularly at 100 mA cm⁻², where the performance gap becomes more pronounced. This further underscores the importance of the mixed RuIr configuration in achieving lower energy requirements at higher current densities.

To provide a clearer comparison of overpotentials at these different current densities for Ru/Ir-based electrocatalysts, we have added Fig. R9 as Fig. 3c in the revised Supplementary Information.

Fig. R9 Comparison of measured overpotential. Overpotentials required to achieve current densities of 10 mA cm⁻² and 100 mA cm⁻² for (RuIr)O₂/C, RuO₂@IrO₂/C, and RuO₂ NPs/C.

Changes Made:

- We have added the Fig. R9 as Fig. 3c on page 37 of the revised Manuscript.
- We have newly added discussion for the overpotential comparison of (RuIr)O₂/C at 10 mA cm⁻² and 100 mA cm⁻² of current densities on page 8 of the revised Manuscript.

Comments 4: Please provide the concentration of Ru and Ir ions in the electrolyte solution after the stability test of different catalysts to demonstrate the reduction of elemental dissolution.

Author's response: We sincerely appreciate the reviewer's insightful suggestion regarding the dissolution behavior of Ru and Ir ions during OER operation. In response, we have conducted an ICP-MS analysis to evaluate the dissolution of Ru and Ir ions from (RuIr)O₂/C, RuO₂@IrO₂/C, and RuO₂ NPs/C (Fig. R10). The ICP-MS results confirmed that (RuIr)O₂/C exhibited negligible leaching of Ru and Ir, demonstrating its highly stable structure during long-term OER operation. In contrast, RuO₂@IrO₂/C and RuO₂ NPs/C showed significant loss of Ru and Ir during OER. These findings are consistent with the elemental mapping (Supplementary Figs. 25, 26) and HRTEM (Supplementary Fig. 27) analysis results, which further corroborate the stability of the (RuIr)O₂/C electrocatalyst.

To clearly illustrate the comparison of Ru and Ir dissolution in each electrocatalyst during OER operation, we have revised Supplementary Fig. 28 in the revised Supplementary Information.

Fig. R10 ICP-MS analysis during OER operation. Leached out **a** Ru and **b** Ir concentrations in the electrolyte after CP test, determined through ICP-MS analysis.

Changes Made:

- We have replaced Supplementary Fig. 28 with Fig. R10 on page 31 of the revised Supplementary Information.
- We have revised the discussion for the ICP-MS analysis results on page 10, line 19 of the revised Manuscript.

Comments 5: Whether catalyst detachment and dissolution occurs during PEMWE testing, especially high-current testing, please identify.

Author's response: We greatly appreciate the reviewer's constructive comment. To address the concern regarding catalyst detachment and dissolution during PEMWE testing, particularly under high current densities, we conducted a detailed analysis using SEM imaging to examine the surface morphology and cross-section of the MEA before and after long-term single cell test. As shown in Figs. R11a,e and R12a,b, surface cracks were observed on all MEAs both before and after durability test, which were attributed to the hot-pressing process.

Notably, the commercial IrO_2 (with a reduction rate of 40.37%) exhibited significant changes in catalyst layer thickness (Fig. R11), whereas the $(\text{RuIr})\text{O}_2/\text{C}$ showed only a modest reduction of about 28.66% during the single-cell durability test (Fig. R12). Moreover, the MEA with the $(\text{RuIr})\text{O}_2/\text{C}$ maintained nearly stable voltage and HFR values throughout the testing period, as shown in Fig. 4e. This suggests that the $(\text{RuIr})\text{O}_2/\text{C}$ remained robust, with minimal degradation, effectively preserving its OER performance during prolonged PEMWE operation. To further demonstrate the stability of the $(\text{RuIr})\text{O}_2/\text{C}$ during single-cell operation, we have included the experimental results along with a detailed discussion in the revised Supplementary Information and Manuscript.

Fig. R11 Structural characterization of commercial IrO_2 electrocatalysts during single-cell test. SEM images for surface morphology **a** before and **b** after single cell durability test. **b,f** Cross section images and

corresponding elemental mapping images for **c,g** iridium (red), and **d,h** fluorine (gray) before and after single-cell durability test.

Fig. R12 Structural characterization of (RuIr)O₂/C electrocatalysts during single-cell test. SEM images for surface morphology **a** before and **b** after single-cell durability test. **c,g** Cross section images and corresponding elemental mapping images for **d,h** fluorine (gray), **e,i** ruthenium (cyan), and **f,j** iridium (red) before and after single-cell durability test.

Changes Made:

- We have newly added the Figs. R11-R12 as Supplementary Figs. 35-36 on page 38-39 of the revised Supplementary Information.
- We have newly added detailed discussion on page 12, line 19 of the revised Manuscript.